# SUN2 mediates calcium-triggered nuclear actin polymerization to cluster active RNA polymerase II

Svenja Ulferts [1✉] & Robert Grosse [1,2✉]

## Abstract

The nucleoskeleton is essential for nuclear architecture as well as genome integrity and gene expression. In addition to lamins, titin or spectrins, dynamic actin filament polymerization has emerged as a potential intranuclear structural element but its functions are less well explored. Here we found that calcium elevations trigger rapid nuclear actin assembly requiring the nuclear membrane protein SUN2 independently of its function as a component of the LINC complex. Instead, SUN2 colocalized and associated with the formin and actin nucleator INF2 in the nuclear envelope in a calcium-regulated manner. Moreover, SUN2 is required for active RNA polymerase II (RNA Pol II) clustering in response to calcium elevations. Thus, our data uncover a SUN2-formin module linking the nuclear envelope to intranuclear actin assembly to promote signal-dependent spatial reorganization of active RNA Pol II.

**Keywords** SUN2; Nuclear Actin; RNA Pol II Clustering; Formin
**Subject Categories** Cell Adhesion, Polarity & Cytoskeleton; Membranes & Trafficking

## Introduction

The mammalian genome is organized, contained and maintained inside the nuclear compartment enclosed by the nuclear envelope (NE) consisting of hundreds to tens of thousands nuclear pore complexes depending on cell type, differentiation and cell cycle stage (McCloskey et al, 2018). Nuclear architecture further critically depends on the nucleoskeleton that connects the chromatin to the inner nuclear membrane (INM) (Adam, 2017), which consist of a variety of inner nuclear transmembrane proteins such as Emerin, Lamin B receptor (LBR), and the Sad1p, Unc-84 (SUN)-domain proteins SUN1 and SUN2 (Pawar and Kutay, 2020). In the perinuclear space, SUN proteins interact with KASH (Klarsicht/ ANC-1/Syne-1 homology)-domain proteins (nesprins), which reside in the outer nuclear membrane (ONM), forming the nuclear envelope (NE) spanning linker of nucleoskeleton and cytoskeleton (LINC) complex (Crisp et al, 2006). At the cytosolic side, nesprins bind to components of the cytoskeleton such as microtubules or actin filaments thus allowing for the direct

propagation of mechanical stimuli towards the nuclear compartment thereby mediating mechanotransduction (Kirby and Lammerding, 2018; Chambliss et al, 2013). However, the LINC complex serves several other roles including nuclear anchorage, insertion of NPCs into the NE, and tethering the telomeres of meiotic chromosomes to the NE (Rothballer and Kutay, 2013; Rothballer et al, 2013).

Recently, dynamic actin filament assembly inside the nucleus has emerged as a more novel part of the nucleoskeleton (Ulferts et al, 2024) promoting activation of serum response factor signaling (SRF) (Plessner et al, 2015; Baarlink et al, 2013), transcription (Wei et al, 2020), DNA repair (Caridi et al, 2018; Schrank et al, 2018), androgen signaling (Knerr et al, 2023), as well as spatial chromatin reorganization (Zagelbaum et al, 2023). We previously showed that G-protein-coupled receptors (GPCRs) and nuclear calcium signaling mediate the rapid formation of nuclear actin filaments, leading to chromatin reorganization towards a more dynamic and open state through the actin regulator inverted formin 2 (INF2) (Wang et al, 2019). INF2 belongs to the formin family of actin nucleators and has been associated with various diseases such as focal segmental glomerulosclerosis (FSGS) and the neurological disorder Charcot-Marie-Tooth (CMT) disease (Zhao et al, 2022; Labat-de-Hoz and Alonso, 2020). All of the more than 70 identified pathogenic INF2 mutations localize to the exons encoding the N-terminal Diaphanous inhibitory domain (DID), where they potentially alter its binding to the Diaphanous autoregulatory domain (DAD) (Labat-de-hoz and Alonso, 2021; Labat-de-Hoz and Alonso, 2020). INF2 has been described to exist as two splice variants: the prenylated INF2-CAAX and the non-prenylated INF2-nonCAAX (Ramabhadran et al, 2011). These isoforms can be expressed in a cell-type-specific manner, with the nonCAAX variant being predominant in U2OS, HeLa, and Jurkat cells (Ramabhadran et al, 2011). Notably, in NIH3T3 cells INF2 mostly exists in a CAAX-box modified isoform that can localize to the ER membrane (Chhabra et al, 2009). In addition to that, INF2 has been reported to be regulated via calcium-calmodulin interactions (Wang et al, 2019; Wales et al, 2016; Bayraktar et al, 2020), whereas a cyclase-associated protein/acetylated actin complex can mediate INF2 autoinhibition (A et al, 2019; Mu et al, 2020). In addition, INF2 has been reported to form heterodimers with mDia formins (Sun et al, 2011).

We previously showed that in addition to the ER, INF2 also resides at the INM and that calcium-triggered transient nuclear actin filament polymerization appears to emanate from the INM (Wang et al, 2019). However, which membrane proteins that localize to the INM, such as

[1]Institute of Experimental and Clinical Pharmacology and Toxicology, Medical Faculty, University of Freiburg, Freiburg, Germany. [2]Centre for Integrative Biological Signalling Studies—CIBSS, University of Freiburg, Freiburg, Germany. ✉E-mail: svenja.ulferts@pharmakol.uni-freiburg.de; robert.grosse@pharmakol.uni-freiburg.de

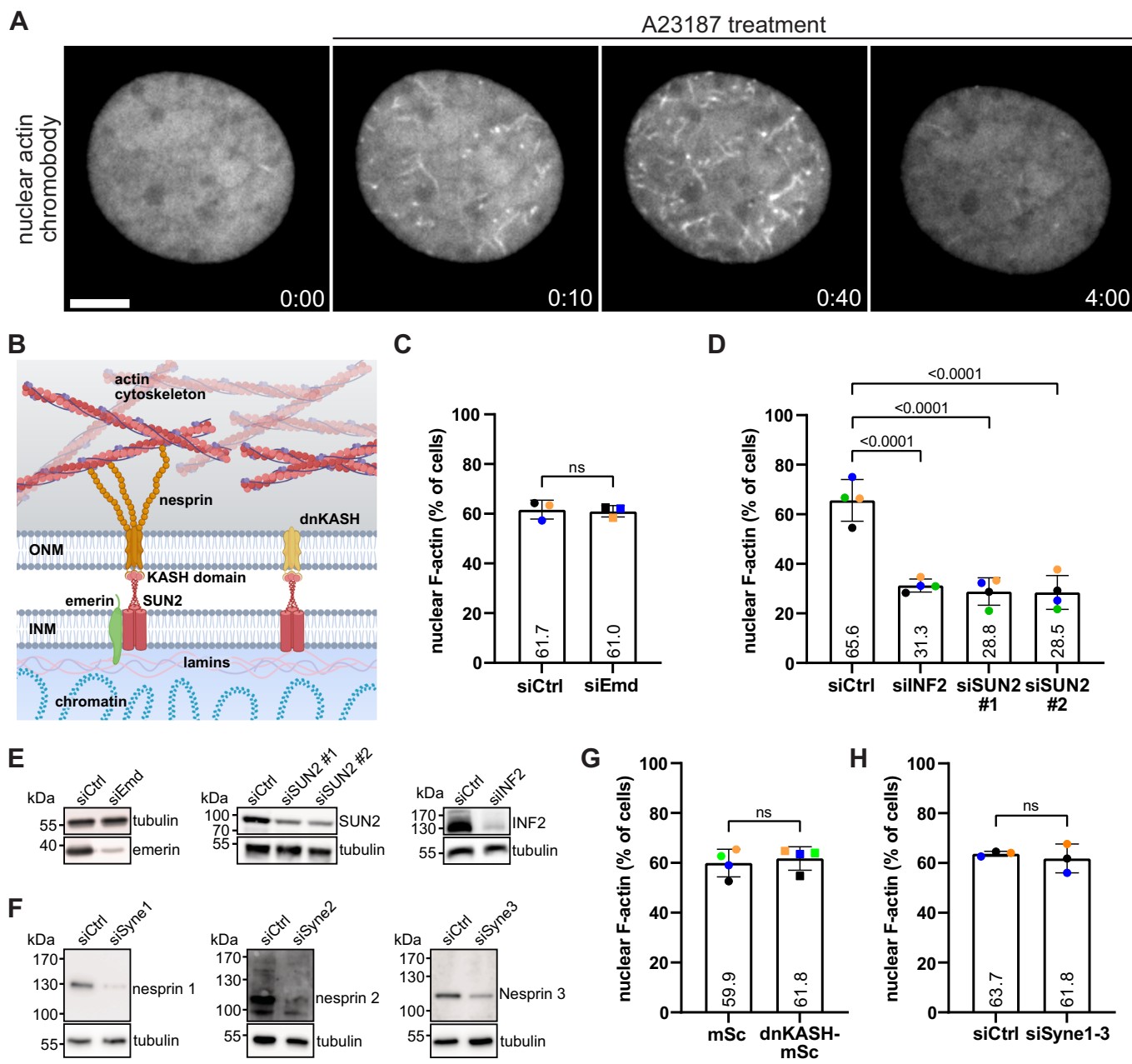

**A** A23187 treatment

nuclear actin chromobody

0:00    0:10    0:40    4:00

**B**
actin cytoskeleton
nesprin
dnKASH
ONM
KASH domain
emerin
SUN2
INM
lamins
chromatin

**C** nuclear F-actin (% of cells)
ns
61.7    61.0
siCtrl    siEmd

**D** nuclear F-actin (% of cells)
<0.0001
<0.0001
<0.0001
65.6    31.3    28.8    28.5
siCtrl    siINF2    siSUN2 #1    siSUN2 #2

**E**
kDa    siCtrl    siEmd
55    tubulin
40    emerin

kDa    siCtrl    siSUN2 #1    siSUN2 #2
100
70    SUN2
55    tubulin

kDa    siCtrl    siINF2
170
130    INF2
55    tubulin

**F**
kDa    siCtrl    siSyne1
170
130    nesprin 1
100
55    tubulin

kDa    siCtrl    siSyne2
170
130    nesprin 2
100
55    tubulin

kDa    siCtrl    siSyne3
170
130    Nesprin 3
100
55    tubulin

**G** nuclear F-actin (% of cells)
ns
59.9    61.8
mSc    dnKASH-mSc

**H** nuclear F-actin (% of cells)
ns
63.7    61.8
siCtrl    siSyne1-3

Lamin B receptor, emerin, LAP2, among others, may be involved in calcium/INF2-mediated nuclear actin assembly and what specific functions are exerted was so far unknown. Here we show that calcium signaling promotes regulation of RNA Pol II activity through the nuclear membrane receptor SUN2 in complex with the formin INF2 (Labat-de-Hoz and Alonso, 2020) to spatially cluster RNA polymerase II (RNA Pol II) in a actin polymerization-dependent manner.

## Results

We previously reported a role for the LINC complex for F-actin formation during cell spreading, which involved nuclear mDia to activate SRF (Plessner et al, 2015). We therefore asked whether

components of the LINC complex are also implicated in nuclear actin assembly in response to rapid calcium elevations that promote the formation of highly transient F-actin structures reorganizing throughout the nucleus before disassembly after 2–4 min (Fig. 1A; Movie EV1). To this end, we silenced the INM proteins emerin and SUN2 in NIH3T3 cells stably expressing the nuclear actin chromobody (nAC) (Plessner et al, 2015) and stimulated them with the calcium ionophore A23187 to trigger the dynamic assembly of nuclear F-actin (Wang et al, 2019). We found that SUN2, like INF2, but not emerin, was critically required for calcium-mediated nuclear actin polymerization (Fig. 1B–E; Movies EV2 and EV3). To test for a potential mechanotransduction role involving the LINC complex we overexpressed a dominant-negative variant of the KASH domain (dnKASH) which saturates

**Figure 1.  Calcium transients trigger INF2 and SUN2-dependent nuclear actin assembly.**

(A) Representative spinning disc confocal slices of a NIH3T3 cell stably expressing nAC-tagGFP showing rapid nuclear F-actin assembly upon addition of 1 μM A23187. Scale bar = 5 μm. (min:sec after drug treatment). (B) Model of the nuclear envelope (NE) spanning linker of nucleoskeleton and cytoskeleton (LINC) complex. It consists of SUN proteins (red) located in the inner nuclear membrane (INM) which bind to the KASH domain of nesprin proteins (orange) residing in the outer nuclear membrane (ONM). In the cytosol, nesprins bind to components of the cytoskeleton including F-actin. Inside the nuclear compartment, SUN proteins interact with nuclear lamins and other inner nuclear membrane proteins like emerin (green). The integrity and mechanotransduction function of the LINC complex can be disrupted by introducing dominant-negative KASH (dnKASH) domains which displace endogenous nesprins from the NE. (C) NIH3T3 cells stably expressing nAC-tagGFP that were transfected with non-targeting control siRNA (siCtrl, n = 611) or siRNA targeting emerin (siEmd, n = 528) were stimulated with 1 μM A23187 and scored for the occurrence of nuclear actin assembly. Average percentage of positive events from n = 3 independent color-coded experiments is shown. Two-tailed t test. ns, not significant. (D) NIH3T3 cells stably expressing nAC-tagGFP were transfected with siCtrl or siRNA targeting INF2 or SUN2 and stimulated with 0.4 U/mL thrombin to induce rapid nuclear actin assembly. In total 453 (siCtrl), 417 (siINF2), 390 (siSUN2 #1) or 500 (siSUN2 #2) cells were scored from n = 4 independent color-coded experiments. One-way ANOVA with Dunnett's multiple comparison test, P < 0.0001. (E) Immunoblots showing siRNA-mediated emerin, SUN2, or INF2 knockdown efficiency. (F) Immunoblots showing siRNA-mediated nesprin1, nesprin2, or nesprin3 knockdown efficiency. (G) NIH3T3 cells stably expressing nAC-tagGFP were transfected with mScarlet-tagged dominant-negative KASH (dnKASH-mSc) or mScarlet (mSc) empty vector and stimulated with 1 μM A23187 to induce rapid nuclear actin assembly. In total, 415 (mSc) or 441 (dnKASH-mSc) cells were scored from n = 4 independent experiments. Mann–Whitney test. ns not significant. (H) NIH3T3 cells stably expressing nAC-tagGFP were transfected with siCtrl or siRNA targeting nesprin1, nesprin2, and nesprin3 and stimulated with 1 μM A23187 to induce rapid nuclear actin assembly. In total, 267 (siCtrl) or 213 (siSyne1-3) cells were scored from n = 3 independent experiments. Two-tailed t test. ns not significant. Data information: In (C, D, G, H), data are represented as mean ± SD. Source data are available online for this figure.

available binding sites of endogenous SUN proteins at the NE thereby uncoupling SUN2 from nesprin-cytoskeletal interactions (Lombardi et al, 2011; Baarlink et al, 2017) (Figs. 1B and EV1A,B). Notably, the expression of dominant-negative KASH did not interfere with calcium-mediated nuclear F-actin (Figs. 1G and EV2). We then depleted the cells for nesprin1, nesprin2 and nesprin3, which again had no effect on A23187 induced nuclear actin assembly (Fig. 1F,H), further indicating that SUN2 functions independently from the LINC complex in this signaling process.

Since SUN2 as well as INF2 localize to the INM we performed structured illumination microscopy (SIM) to investigate their precise distribution at the nuclear envelope at nanoscale resolution. Endogenous INF2 was detectable at the nuclear membrane as identified by Lamin A/C co-staining (Fig. 2A). Interestingly, we observed distinct colocalizations of endogenous INF2 with endogenous SUN2, specifically at the nuclear rim, suggesting that both proteins can be in close vicinity at the INM (Figs. 2B and EV3A,B). The mean fluorescence intensity of INF2 at the nuclear membrane increased within 60 s after calcium ionophore addition (Fig. 2C), suggesting a rather dynamic alteration in the local distribution of INF2 on the same timescale as induced actin polymerization. To test whether SUN2 and INF2 can interact in vitro, we performed co-immunoprecipitation experiments in HEK293T cells. This revealed that INF2 coprecipitated with SUN2-FLAG in the presence of calcium, supporting the notion that both proteins physically associate (Fig. EV3C).

To further corroborate a potential close interrelationship between SUN2 and INF2 we turned to proximity ligation assays (PLA), which enable the study of endogenous protein colocalizations below 40 nm using specific primary antibodies targeting the two proteins of interest. Notably, INF2 and SUN2 robustly and significantly colocalized during PLAs and this association was further increased in response to calcium elevation using the calcium ionophore A23187 (Fig. 3A,B). Consistently, the majority of PLA foci was found to be enriched at the nuclear periphery (Fig. 3C, and EV4A,B), indicating that SUN2 and INF2 associate at the NE. As may be expected, pre-treatment of the cells with the calcium chelator BAPTA-AM resulted in a decrease of detected PLA foci even below the number observed in untreated cells (Figs. 3D and EV4C). These data demonstrate that endogenous INF2 and SUN2 dynamically associate in a calcium-dependent manner

inside the nucleus. To confirm specificity of the detected PLA signal, we performed control experiments by omitting each primary antibody separately or both together (Figs. 3A and EV4D,E) or as a positive control we probed for Lamin A/C or emerin as known interactors of SUN2 (Fig. EV4D,F). Additional negative controls were included using primary antibodies for SUN2 together with the cytosolic protein tubulin (Fig. EV4D) or siSUN2-mediated downregulation prior to fixation for SUN2 antibody specificity (Fig. EV4D,G).

We next tested whether the observed changes in F-actin formation have an implication on inducible transcription. During the transcription initiation process, RNA Pol II first becomes phosphorylated at serine 5 (Pol2pS5) of the carboxy-terminal domain (CTD) repeat YSPTSPS (Phatnani and Greenleaf, 2006). Therefore, we decided to investigate the spatial distribution of endogenous Pol2pS5 and noticed the assembly of active RNA Pol II cluster of different morphologies upon A23187 treatment. Since calcium-induced nuclear F-actin formation is a fast process that occurs within seconds to minutes, we wondered how quickly these clusters form and for how long they persist. We thus fixed and stained the cells at different time points between 0 and 120 min after nuclear actin assembly and quantified their size and number in control cells (Fig. 4A,C–E) as compared to cells depleted for SUN2 (Fig. 4B,F–H). Notably, the average volume of Pol2pS5 foci significantly and robustly increased after induction of nuclear actin assembly (Fig. 4A,C). Strikingly, the formation of large clusters with a volume ≥ 0.1 μm$^3$ could be observed as soon as 5 min post A23187 treatment (Fig. 4A,D,E). We then depleted SUN2 to test, whether the observed calcium-induced clustering of initiating RNA Pol II is SUN2-dependent and found that none of the changes observed over time in control cells could be detected in SUN2-depleted cells (Fig. 4B,F–H).

To specifically address the role of polymerized nuclear actin, we either depleted the cells for the responsible actin nucleator INF2 (Fig. 5A–H), or overexpressed a nuclear-targeted version of the non-polymerizable actin mutant R62D (myc-NLS-mScarlet-Actin$^{R62D}$) (Fig. 6A–H) (Posern et al, 2002). Both approaches interfering with nuclear F-actin assembly led to a reduction in RNA Pol II cluster formation upon calcium ionophore treatment (Figs. 5B,F–H and 6B,F–H), comparable to the effect of SUN2 silencing (Fig. 4). Together these results demonstrate that the rapid and transient INF2/SUN2-dependent assembly and disassembly of nuclear actin filaments drives clustering of active RNA Pol II.

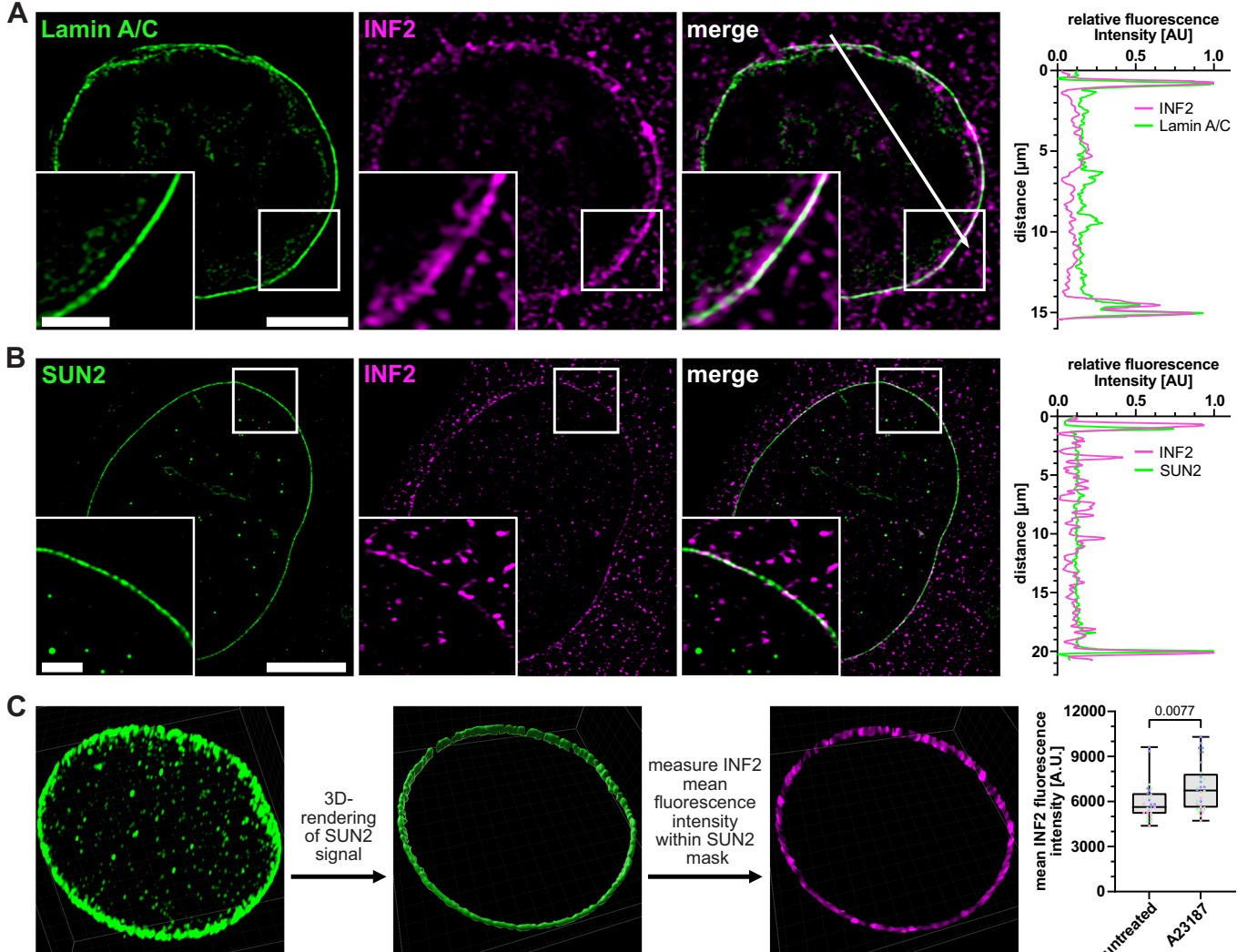

**Figure 2. SUN2 and INF2 associate at the inner nuclear membrane.**

(A) Immunofluorescence (IF) staining of endogenous Lamin A/C (green) and INF2 (magenta) in a NIH3T3 cell. Images show a representative optical z-slice acquired by structured illumination microscopy (SIM) imaging. Box indicates magnified area of individual SIM optical z-slice shown in the lower left corner of each panel. Scale bar = 5 μm (overview) or 2 μm (zoom). A corresponding line-scan profile was added in the right panel. Line-scan shows normalized fluorescence intensity of Lamin A/C (green) and INF2 (magenta) as measured along the white arrow. (B) IF of endogenous SUN2 (green) and INF2 (magenta) in NIH3T3 cells shows colocalization of both proteins at the nuclear membrane. Box indicates magnified area of individual SIM optical z-slice shown in the lower left corner of each panel. Scale bar = 10 μm (overview) or 2 μm (zoom). A corresponding line-scan profile was added in the right panel. Line-scan shows the normalized fluorescence intensity of SUN2 (green) and INF2 (magenta) as measured along the white arrow. (C) Overview of the work flow to measure mean fluorescence intensity of INF2 (magenta) within the 3D-rendered mask created based on SUN2 (green) fluorescence intensity (left). NIH3T3 cells were either untreated or treated with 1 μM A23187 before being co-stained for SUN2 and INF2. The mean fluorescence intensity of INF2 was then measured within the created SUN2 mask. Data are presented as box plots with individual data points from n = 3 independent experiments (right). The boxes indicate the first and third quartile, with the median shown as horizontal line. Whiskers extend from the minimum to maximum value. Mann–Whitney test. Source data are available online for this figure.

## Discussion

Nuclear actin dynamics have emerged as a crucial regulator of nuclear functions including transcriptional control by driving androgen receptor condensate formation (Knerr et al, 2023) or promoting RNA Pol II clustering in response to serum stimulation (Wei et al, 2020). In this study, we identify the inner nuclear membrane protein SUN2 as a key player in the rapid assembly of a highly dynamic nuclear actin filament network triggered by calcium signaling within seconds. Intriguingly, this function appears to be independent of the mechanotransduction function associated with the SUN2-containing LINC complex as both the overexpression of a dominant-negative KASH mutant and the knockdown of endogenous nesprin proteins had no impact on calcium-mediated nuclear actin assembly. Ultimately, this rapid response resulted in a SUN2- and nuclear F-actin-dependent increase in volume of active RNA Pol II clusters, revealing a specific role in transcriptional regulation.

We had recently shown that intranuclear calcium elevations precede rapid and transient assembly of nuclear actin filaments (Safaralizade et al, 2021; Wang et al, 2019). These F-actin structures appear to be polymerized independently of Arp2/3 but require the formin INF2

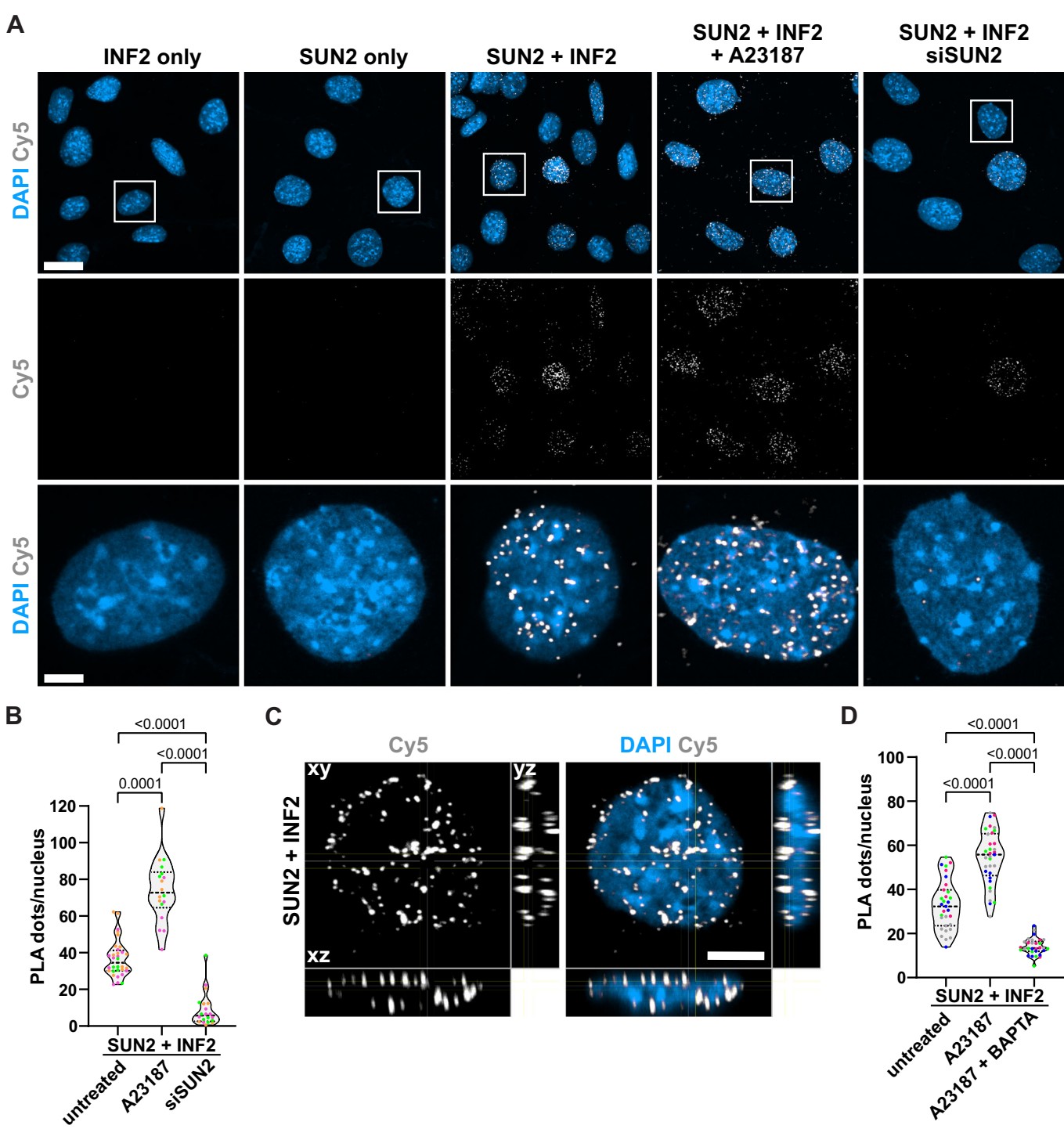

leading to a decrease in heterochromatin and increased chromatin mobility (Safaralizade et al, 2021; Wang et al, 2019). Here we identified a complex at the INM consisting of SUN2 and INF2. Our data support the close vicinity of these two proteins. However, we do not conclude that they physically interact. Interestingly, both INF2 and SUN2 have been reported to be directly regulated by calcium elevations with INF2 binding calcium calmodulin (CaM) to activate this formin (Wales et al, 2016; Wang et al, 2019) and SUN2 switching between a nesprin-binding

trimeric and monomeric state with higher calcium concentrations favoring monomeric SUN2 not involved in LINC complex formation (Jahed et al, 2018). The LINC complex plays a role in coupling mechanics to chromatin-associated processes (Pawar and Kutay, 2020). Applying shear stress to the surface of mammalian cells results in SUN protein-dependent chromatin deformation and upregulation of transcription within minutes of force application (Tajik et al, 2016). Emerin, a known nucleoplasmic binding partner of SUN2 (Haque et al, 2010),

**Figure 3. INF2 and SUN2 dynamically associate in a calcium-dependent manner at the NE.**

(A) NIH3T3 cells were either untreated, stimulated with 1 μM A23187 for 1 min or transfected with SUN2-targeted siRNA for 72 h before being subjected to PLA using the indicated primary antibodies. Individual antibodies serve as negative control. Images show representative maximum-intensity projections of 20 confocal z-slices (0.21 μm z-distance). PLA signals (Cy5) indicating the interaction of endogenous target proteins within the nucleus (DAPI) were detected as discrete spots. The white box indicates the magnified area shown in the lower panel. Scale bar = 10 μm (overview) or 2 μm (zoom). (B) Quantification of PLA signal (dots/nucleus) shown as violin plot with individual data points, median and quartiles. Individual data points represent mean from one field of view acquired from three independent color-coded biological replicates. Fields of view imaged: control $n = 32$, A23187 $n = 18$, siSUN2 $n = 20$. One-way ANOVA with Kruskal–Wallis multiple comparison test, $P$ (untreated versus A23187) = 0.0001, $P$ (untreated vs. siSUN2) < 0.0001, $P$ (A23187 vs. siSUN2) < 0.0001. (C) Orthogonal optical cross section reconstructed from a Z-scan through a NIH3T3 cell nucleus stained for DAPI and PLA dots (Cy5) between INF2 and SUN2. Scale bar = 5 μm. (D) Quantification of PLA signal in cells pre-treated with BAPTA-AM shown as violin plot with individual data points, median and quartiles. Data points represent individual values from four independent color-coded biological replicates. Fields of view imaged: control $n = 31$, A23187 $n = 33$, A23187 + BAPTA $n = 30$. One-way ANOVA with Tukey´s multiple comparison test, $P < 0.0001$. Source data are available online for this figure.

also associates with lamin A/C, actin and myosins (Holaska et al, 2004; Holaska and Wilson, 2007; Pawar and Kutay, 2020), making it perfectly positioned to integrate mechanical stimuli at the INM. However, neither the knockdown of emerin or nesprins 1, 2, and 3, nor the overexpression of dnKASH, which uncouples SUN2 from its mechanotransduction function as part of the LINC complex, had an effect on rapid and transient nuclear F-actin assembly. We thus propose that a SUN2/INF2 complex operates independently from mechanotransduction to relay rapid calcium signaling to nuclear actin assembly. Notably, INF2 has previously been described to elicit a rapid and transient actin reset in response to increased intracellular calcium levels, a process termed Calcium-mediated actin reset (CaAR) (Wales et al, 2016) and the INF2-dependent formation of a perinuclear actin rim is mediated by calcium (Shao et al, 2015). We found that at least a part of the cellular INF2 pool redistributed to the nuclear membrane upon A23187 treatment, and that SUN2 and INF2 associate in a calcium-dependent manner. We hence hypothesize that activated INF2 interacts with monomeric SUN2, potentially assisting in maintaining the open conformation of this formin. Active INF2 then rapidly polymerizes nuclear actin into long filaments, leading to a more open and dynamic chromatin state that facilitates RNA Pol II cluster formation (Fig. 7).

Actin has been described as a component of chromatin remodeling complexes (Kapoor and Shen, 2014) as well as of RNA polymerases (Hu et al, 2004; Hofmann et al, 2004; Philimonenko et al, 2004), implicating its involvement in transcriptional regulation. Conversely, dynamically polymerized actin, has only recently emerged in transcription control processes: in MRTF-A/SRF-mediated gene transcription (Baarlink et al, 2013; Plessner et al, 2015), downstream of T-cell receptor activation (Tsopoulidis et al, 2019), or in response to androgen signaling (Knerr et al, 2023), among others. It is well established that active RNA Pol II organizes into distinct nuclear foci often termed transcription factories (Rippe and Papantonis, 2022). These foci form transiently and stimuli affecting gene transcription alter their dynamics, implying that clustering is a regulated process that allows cells to rapidly respond to external signals (Cisse et al, 2013). In unstimulated cells, the presence of stable nuclear actin filaments correlates with impaired RNA Pol II localization and global transcription (Serebryannyy et al, 2016). However, stimulation of cells with serum or interferon-γ induces the formation of RNA Pol II clusters, presumably in the transcription initiation phase. This phenomenon has been observed through imaging of fluorescently labeled RPB1, the catalytic subunit of RNA Pol II, 30 min after serum stimulation. Even though the authors did not clearly show the dynamic assembly of nuclear actin filaments, serum-enhanced RNA Pol II cluster formation was impaired upon overexpression of nuclear-localized actin mutants that perturb the polymerization state of nuclear actin (Wei et al, 2020). Both of these studies argue for the requirement of a dynamic pool of nuclear actin.

Given the similar rapid response observed in the nucleus upon calcium elevations, we found that dynamic actin polymerization impacts on spatial organization of active RNA Pol II within minutes of stimulation. These findings implicate a direct correlation between the visualization of dynamic nuclear actin assembly and signal-induced clustering of active RNA Pol II, shedding new light on transcriptional regulation mechanisms. Previous work suggested a role for the actin assembly factors N-WASP and Arp2/3 in RNA Pol II clustering and transcription (Yoo et al, 2007; Wei et al, 2020), while our data favor a mechanism by INF2, at least in the context of calcium signaling (Safaralizade et al, 2021; Wang et al, 2019). We propose the signal-mediated formation of a SUN2/INF2 complex at the INM that regulates nuclear actin filament assembly supporting the notion that nuclear F-actin-dependent formation of transcription factories may represent a more general mechanism.

## Methods

### Reagents and tools table

| Reagent/resource | Reference or source | Identifier or catalog number |
|---|---|---|
| **Experimental models** | | |
| NIH3T3 cells (*M. musculus*) | ATCC | CRL-1658 |
| NIH3T3-nAC-tagGFP (*M. musculus*) | Grosse Lab | N/A |
| HEK293T cells (*H. sapiens*) | ATCC | CRL-3216 |
| **Recombinant DNA** | | |
| pEF-mScarlet | This study | N/A |
| pEF-mScarlet-dnKASH | This study | N/A |
| pEF-hINF2-CAAX | This study | N/A |
| pEF-hSUN2-FLAG | This study | N/A |
| pEF-myc-NLS-mScarlet | This study | N/A |
| pEF-myc-NLS-mScarlet-Actin$^{R62D}$ | This study | N/A |
| **Antibodies** | | |
| Rabbit anti-tubulin | Cell Signaling Technology | 2125 |
| Rabbit anti-emerin | Cell Signaling Technology | 30853 |
| Rabbit anti-INF2 | ProteinTech | 20466-1-AP |
| Rabbit anti-SUN2 | Abcam | ab124916 |
| Rabbit anti-FLAG(DYKDDDDK) tag | Cell Signaling Technology | 14793 |
| Mouse anti-Syne3 | OriGene | AM33013PU-N |
| Rabbit anti-Syne2 | Invitrogen | PA5-78438 |
| Rabbit anti-Syne1 | Abcam | ab192234 |
| Anti-Rabbit IgG, HRP linked | Cell Signaling Technology | 7074 |

| Reagent/resource | Reference or source | Identifier or catalog number |
|---|---|---|
| Anti-Mouse IgG, HRP linked | Rockland | 310-703-002 |
| Mouse anti-SUN2 | NovusBio | NBP2-59942 |
| Rabbit anti-Lamin A/C | ProteinTech | 10298-1-AP |
| Mouse anti-Lamin A/C | Cell Signaling Technology | 4777 |
| Rat anti-RNA Pol II CTD S5ph | Active Motif | 61701 |
| Alexa Fluor 647 goat anti-rabbit IgG | Invitrogen | A21244 |
| Alexa Fluor 488 chicken anti-mouse IgG | Invitrogen | A21200 |
| Alexa Fluor 488 goat anti-rat IgG | Invitrogen | A11006 |
| Alexa Fluor 488 goat anti-mouse IgG | Invitrogen | A11029 |
| **Oligonucleotides and other sequence-based reagents** | | |
| AllStars Negative Control siRNA | Qiagen, no. 1027281 | AATTCTCCGAACGTGTCACGT |
| siINF2 | Qiagen, no. SI00822031 | TAGGCTCTAGGGAACAAATAA |
| siSUN2 #1 | Qiagen, no. SI00912765 | CACGTAGAACTCCCTGCATAA |
| siSUN2 #2 | Qiagen, no. SI00912772 | CCGGTTAGTGTTCGGGTGAAA |
| Negative control siRNA | OriGene, no. SR3004 | CGUUAAUCGCGUAUAAUACGCGUAT |
| siSyne1 | Thermo Fisher Scientific, no. s234287 | GGAUGGAACUAGAACAUAU |
| siSyne2 | OriGene, no. SR23729C | GAGCUUUUGACUCAUGUACAGUAAA |
| siSyne3 | OriGene, no. SR421519B | AAGCUACGUAGAAUCAUCACAAUGA |
| siEmd | Invitrogen, no. s65469 and no. s65468 | GGCUUAUCAUAUUAUCCUA and CUAUAAUGAUGACUACUAU |
| mSc-dnKASH_fwd | This study | 5′-gttcactagcaacctcaaacagacac catggtgagcaagggcgaggcag-3′ |
| mSc-dnKASH_rev | This study | 5′-gaatttctagactagtctatttcagagtgg aggagggccattcgtgtatc -3′ |
| BB_pEF_fwd | This study | 5′-aatagactagtctagaaattcaccccaccag-3′ |
| BB_pEF_rev | This study | 5′-ggtgtctgtttgaggttgctagtgaacac-3′ |
| Gibson_pEF-hINF2_fwd | This study | 5′-caacctcaaacagacaccatgtcggtgaaggag ggcgcaca-3′ |
| Gibson_pEF-hINF2_rev | This study | 5′-gtggggtgaatttctagactagtttactggatcaca cacagtttcttg-3′ |
| BB_pEF_fwd_2 | This study | 5′-actagtctagaaattcaccccaccagtgcag-3′ |
| BB_pEF_rev_2 | This study | 5′-ggtgtctgtttgaggttgctagtgaacacagttg tgtc-3′ |
| SUN2_gibson_fwd | This study | 5′-gcccgggatccaccggtcatgtcccgaagaagc cagcgc-3′ |
| SUN2-FL_gibson_rev | This study | 5′-gtcatccttgtaatcagacccccgagtgggcgg gctcccc-3′ |
| BB_FLAG_fwd | This study | 5′-gggtctgattacaaggatgacgacgataagtgag-3′ |
| BB_FLAG_rev | This study | 5′-gaccggtggatcccgggccc-3′ |
| pEF-hSUN2_fwd_gibson | This study | 5′-caacctcaaacagacaccatgtcccgaagaagc cagc-3′ |
| pEF-hSUN2-FLAG_rev_gibson | This study | 5′-ggggtgaatttctagactagtctatttcactta tcgtcgtcatccttgt-3′ |
| BB_pEF_fwd | This study | 5′-aatagactagtctagaaattcaccccaccag-3′ |
| BB_pEF_rev | This study | 5′-ggtgtctgtttgaggttgctagtgaacac-3′ |
| **Chemicals, enzymes, and other reagents** | | |
| DMEM High Glucose w/ stable Glutamine w/ Sodium Pyruvate | anprotec | AC-LM-0013 |
| Penicillin/Streptomycin | anprotec | AC-AB-0024 |
| Fetal calf serum | anprotec | AC-SM-0190 |
| Lipofectamine 2000 | Thermo Fisher Scientific | 11668019 |

| Reagent/resource | Reference or source | Identifier or catalog number |
|---|---|---|
| FuGENE® HD Transfection Reagent | Promega | E2311 |
| Lipofectamine RNAiMAX | Thermo Fisher Scientific | 13778150 |
| Bovine Fibronectin | Sigma-Aldrich | F1141 |
| ANTI-FLAG® M2 Affinity Gel | Merck | A2220 |
| Calcium Ionophore A23187 | Sigma-Aldrich | C7522-1MG |
| Thrombin | Sigma-Aldrich | 1.12374.0001 |
| BAPTA-AM | SelleckChem | S7534 |
| Gibson Assembly Master Mix | New England Biolabs | E2611L |
| NcoI-HF | New England Biolabs | R3193L |
| SpeI-HF | New England Biolabs | R3133L |
| T4 DNA Ligase | New England Biolabs | MO202S |
| Q5 High-Fidelity DNA Polymerase | New England Biolabs | M0491L |
| cOmplete™, EDTA-free Protease Inhibitor Cocktail | Roche | 11873580001 |
| Power Blotter 1-Step™ Transfer Buffer (5X) | Invitrogen | PB7300X3 |
| SuperSignal West Femto Maximum Sensitivity Substrate | Thermo Fisher Scientific | 34096 |
| Transfer membrane ROTI®PVDF | Carl Roth | T830.1 |
| ibiTreat® µ-Slide 8 Well | ibidi | 80826 |
| ProLong Diamond Antifade Mountant | Invitrogen | P36961 |
| Duolink® In Situ PLA® Sonde Anti-Maus MINUS | Sigma | DUO92004-100RXN |
| Duolink® In Situ PLA® Probe Anti-Rabbit PLUS | Sigma | DUO92002-100RXN |
| Duolink® In Situ Detection Reagents Farred | Sigma | DUO92013-100RXN |
| DAPI | Sigma-Aldrich | D9542 |
| VenorGem Classic Mycoplasma Detection Kit | Minerva Biolabs | 11-1100 |
| **Software** | | |
| Graph Pad Prism 10.1.0 | https://www.graphpad.com/ | |
| Zen 3.0 Black Edition | Zeiss | |
| ZenBlue | Zeiss | |
| ImageJ/Fiji | https://imagej.net/software/fiji/ | |
| Imaris 10.1.0 | Oxford Instruments | |
| SnapGene | https://www.snapgene.com/ | |
| **Other** | | |
| ELYRA 7 microscope | Zeiss | |
| LSM800 confocal laser-scanning microscope | Zeiss | |
| Yokogawa CSU-X1 spinning disc with Photometrics Prime BSI camera | Yokogawa, Teledyne Photometrics | |
| Power Blotter XL System | Invitrogen | PB0013 |

## Cell culture, transfection, and RNA interference

HEK293T cells (American Type Culture Collection, CRL-3216), NIH3T3 cells (American Type Culture Collection, CRL-1658) and its derivatives (NIH3T3-nAC-tagGFP) were maintained in high-glucose Dulbecco´s modified Eagle´s medium (DMEM, anprotec) supplemented with 10% fetal calf serum (FCS, anprotec), 100 U ml$^{-1}$ penicillin and 100 µg ml$^{-1}$ streptomycin (anprotec) at 37 °C in a 5%

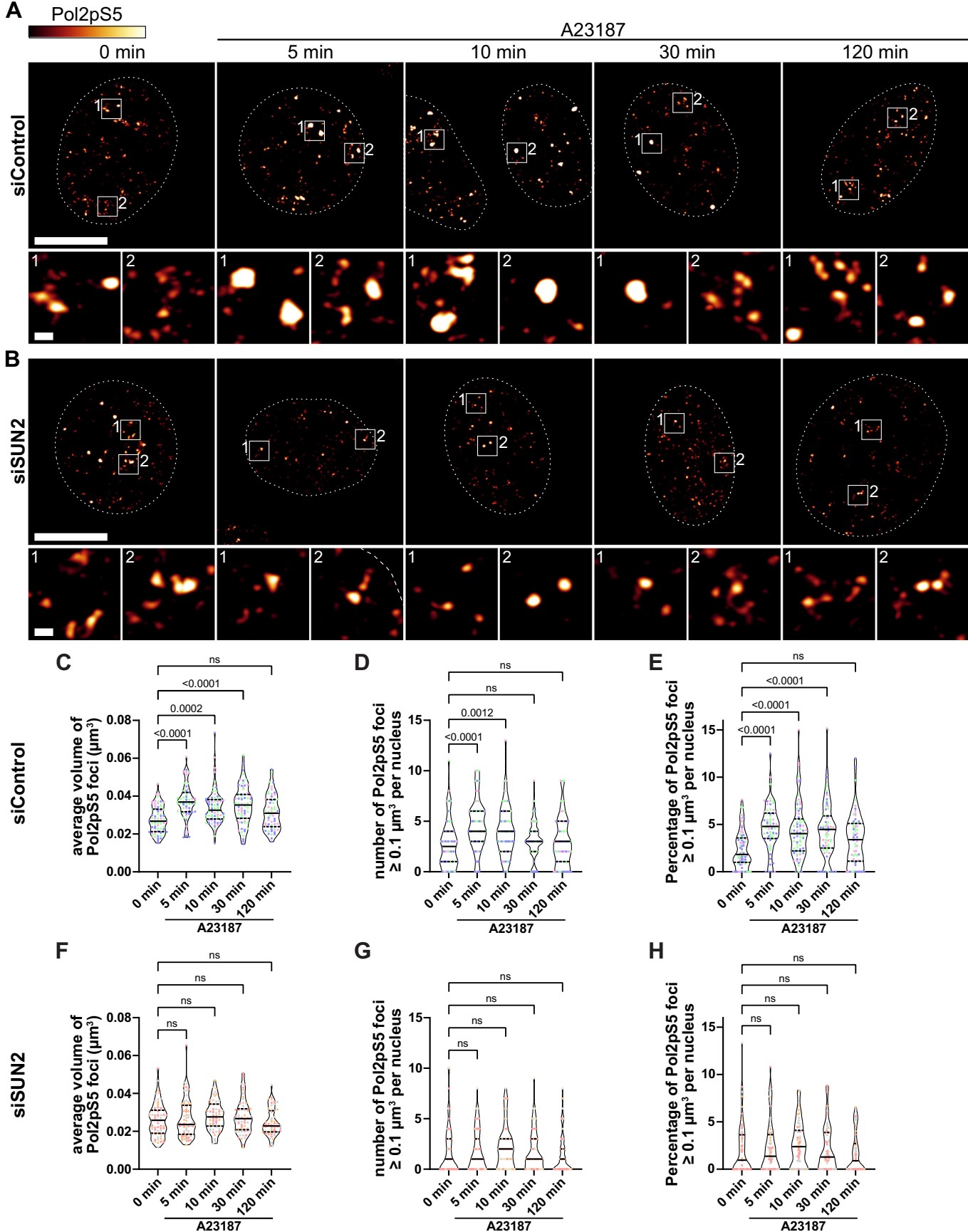

Figure 4. **SUN2-dependent RNA Pol II clustering upon dynamic nuclear actin assembly.**

(A, B) Representative images of IF staining of endogenous phosphorylated RNA Pol II (Pol2pS5) in NIH3T3 control cells (A) or cells depleted for SUN2 (B) and stimulated with 1 μM A23187 for 0, 5, 10, 30, or 120 min prior to fixation. Nuclei are indicated by dashed lines. Gray values of detected fluorescence intensity are color-coded as depicted. White boxes indicated the magnified areas shown below. Scale bar = 10 μm (overview) or 500 nm (zoom). (C–E) Quantification of (C) average volume of detected Pol2pS5 foci (minimal volume of 0.005 μm$^3$ used as cutoff), (D) number of Pol2pS5 foci larger than 0.1 μm$^3$, and (E) percentage of Pol2pS5 foci larger than 0.1 μm$^3$ from $n = 70$, 65, 73, 64 or 60 (0, 5, 10, 30, and 120 min., respectively) siCtrl treated cells from three independent experiments. Data are shown as violin plots with individual data points, median and quartiles. $P$ values were calculated by one-way ANOVA with the Kruskal–Wallis multiple comparison test. Average volume $P$ (0 min vs. 5 min) <0.0001, $P$ (0 min vs. 10 min) = 0.0002, $P$ (0 min vs. 30 min) <0.0001, number of Pol2pS5 foci $P$ (0 min vs. 5 min) <0.0001, $P$ (0 min vs. 10 min) = 0.0012, Percentage of Pol2pS5 foci $P$ (0 min vs. 5 min) <0.0001, $P$ (0 min vs. 10 min) <0.0001, $P$ (0 min vs. 30 min) <0.0001. (F–H) Quantification of (F) average volume of detected Pol2pS5 foci (minimal volume of 0.005 μm$^3$ used as cutoff), (G) number of Pol2pS5 foci larger than 0.1 μm$^3$, and (H) percentage of Pol2pS5 foci larger than 0.1 μm$^3$ from $n = 61$, 67, 58, 58 or 57 (0, 5, 10, 30, and 120 min., respectively) siSUN2 treated cells from three independent experiments. Data are shown as violin plots with individual data points, median and quartiles. Statistical analysis was performed by one-way ANOVA with the Kruskal–Wallis multiple comparison test. n.s. not significant. Source data are available online for this figure.

$CO_2$ atmosphere. All cells were regularly tested for mycoplasma contamination using the VenorGem Classic Mycoplasma detection kit (Minerva Biolabs). Calcium elevations in cells were induced via the addition of the calcium ionophore A23187 (Sigma) as indicated in the respective experiments. Plasmid DNA transfections were conducted using FuGene HD transfection reagent (Promega) according to the supplier's protocol. For siRNA-mediated protein depletion, cells were transfected with targeting or non-targeting siRNA using Lipofectamine RNAiMAX (Invitrogen). Experiments involving SUN2, INF2, nesprin1 (Syne1), nesprin2 (Syne2) and nesprin3 (Syne3) depletion were conducted 72 h post transfection. The utilized targeting sequences and siRNAs were as follows:

Ctrl, AllStars Negative Control siRNA, AATTCTCCGAACGTG TCACGT (Qiagen, no. 1027281);

*Inf2*, TAGGCTCTAGGGAACAAATAA (Qiagen, no. SI00822031);

*Sun2 #1*, CACGTAGAACTCCCTGCATAA (Qiagen, no. SI00912765);

*Sun2 #2*, CCGGTTAGTGTTCGGGTGAAA (Qiagen, no. SI00912772);

*Syne1*, GGAUGGAACUAGAACAUAU (Thermo Fisher Scientific, no. s234287);

*Syne2*, GAGCUUUUGACUCAUGUACAGUAAA (OriGene Techn., no. SR23729C);

*Syne3*, AAGCUACGUAGAAUCAUCACAAUGA (OriGene Techn., no. SR421519B);

Negative control siRNA, CGUUAAUCGCGUAUAAUACGC-GUAT (OriGene Techn., no. SR3004).

For Emd depletion, cells were transfected with a 1:1 mixture of two siRNAs at day 0 and day 2, and subsequent experiments were performed on day 4. The following siRNAs were used:

*Emd*, GGCUUAUCAUAUUAUCCUA and CUAUAAUGAU-GACUACUAU (Invitrogen, no. s65469 and no. s65468).

## Plasmids and reagents

To ensure efficient transcription in NIH3T3 cells, mScarlet-dnKASH (mSc-dnKASH) and mScarlet (mSc) were subcloned into a pEF-backbone vector. mSc-dnKASH was amplified from pCMV-mSc-dnKASH and pEF-mSc-dnKASH was generated using the Gibson cloning technique (NEB) with the following primers (5′→3′): mSc-dnKASH_fwd, gttcactagcaacctcaaacagacaccatggtgag-caagggcgaggcag; mSc-dnKASH_rev, gaatttctagactagtctatttcagagtg-gaggagggccattcgtgtatc. The pEF backbone was amplified using the

following primers (5′→3′): BB_pEF_fwd, aatagactagtctagaaatt-caccccaccag, and BB_pEF_rev, ggtgtctgtttgaggttgctagtgaacac.

The pEF-mScarlet plasmid was generated following standard restriction enzyme cloning procedures. mScarlet was amplified by PCR from pEF-mScarlet-dnKASH introducing restriction sites for NcoI and SpeI using the following primers: mScarlet_NcoI_fwd, acagacaccatggtgagcaag and mScarlet_SpeI_rev: tataatactagtttacttg-tacagctcgtcc. The insert was then ligated into the NcoI-SpeI-digested pEF-mScarlet-dnKASH destination vector.

pEF-hINF2-CAAX was cloned using the Gibson cloning technique. Human INF2-CAAX was amplified from the previously described (Wang et al, 2019) TagBFP2-INF2-CAAX plasmid using the following primers (5′→3′): Gibson_pEF-hINF2_fwd, caacct-caaacagacaccatgtcggtgaaggagggcgcaca; Gibson_pEF-hINF2_rev, gtggggtgaatttctagactagtttactggatcacacacagtttctttg. The insert was then cloned into the pEF-backbone vector, which was amplified from the pEF-mScarlet plasmid using the following primers (5′→3′): BB_pEF_fwd_2, actagtctagaaattcaccccaccagtgcag; BB_pE-F_rev_2, ggtgtctgtttgaggttgctagtgaacacagttgtgtc.

pEF-hSUN2-FLAG was generated using the Gibson cloning technique. First, SUN2 was PCR-amplified from pEGFP-N3-hSUN2 and a C-terminal FLAG-tag added using the following primers (5′→3′): SUN2_gibson_fwd, gcccgggatccaccggtcatgtcccgaagaagccagcgcc; SUN2-FL_gibson_rev, gtcatccttgtaatcagaccccgagtgggcgggctcccc; BB_FLAG_fwd, gggtctgattacaaggatgacgacgataagtgag; BB_FLAG_rev, gaccggtggatcccgggc cc. In the next step, the entire SUN2-FLAG sequence was inserted into a pEF-backbone vector using the following primers (5′→3′): pEF-hSUN2_fwd_gibson, caacctcaaacagacaccatgtcccgaagaagccagc; pEF-hSUN2-FLAG_rev_gibson,ggggtgaatttctagactagtctatttcacttatcgtcgt-catccttgt; BB_pEF_fwd, aatagactagtctagaaattcaccccaccag, and BB_pEF_rev, ggtgtctgtttgaggttgctagtgaacac.

All generated plasmids were sequence-verified.

## Co-immunoprecipitation

All steps were carried out on ice if not indicated otherwise with centrifugation performed at 4 °C. HEK293T cells were seeded on 10-cm$^2$ dishes, transfected with hSUN2-FLAG or INF2-CAAX separately using Lipofectamine 2000 (Thermo Fisher Scientific) according to the supplier's protocol, and harvested by scraping into 1 mL PBS after 24 h. After a washing step with PBS at 3000× $g$ for 5 min, the cells were lysed in Co-IP buffer (20 mM Tris pH 7.5, 150 mM NaCl, 5 mM $CaCl_2$, 1% glycerol, 1% NP-40, 0.05% SDS, 1× cOmplete protease inhibitor cocktail EDTA-free (Roche)) and the

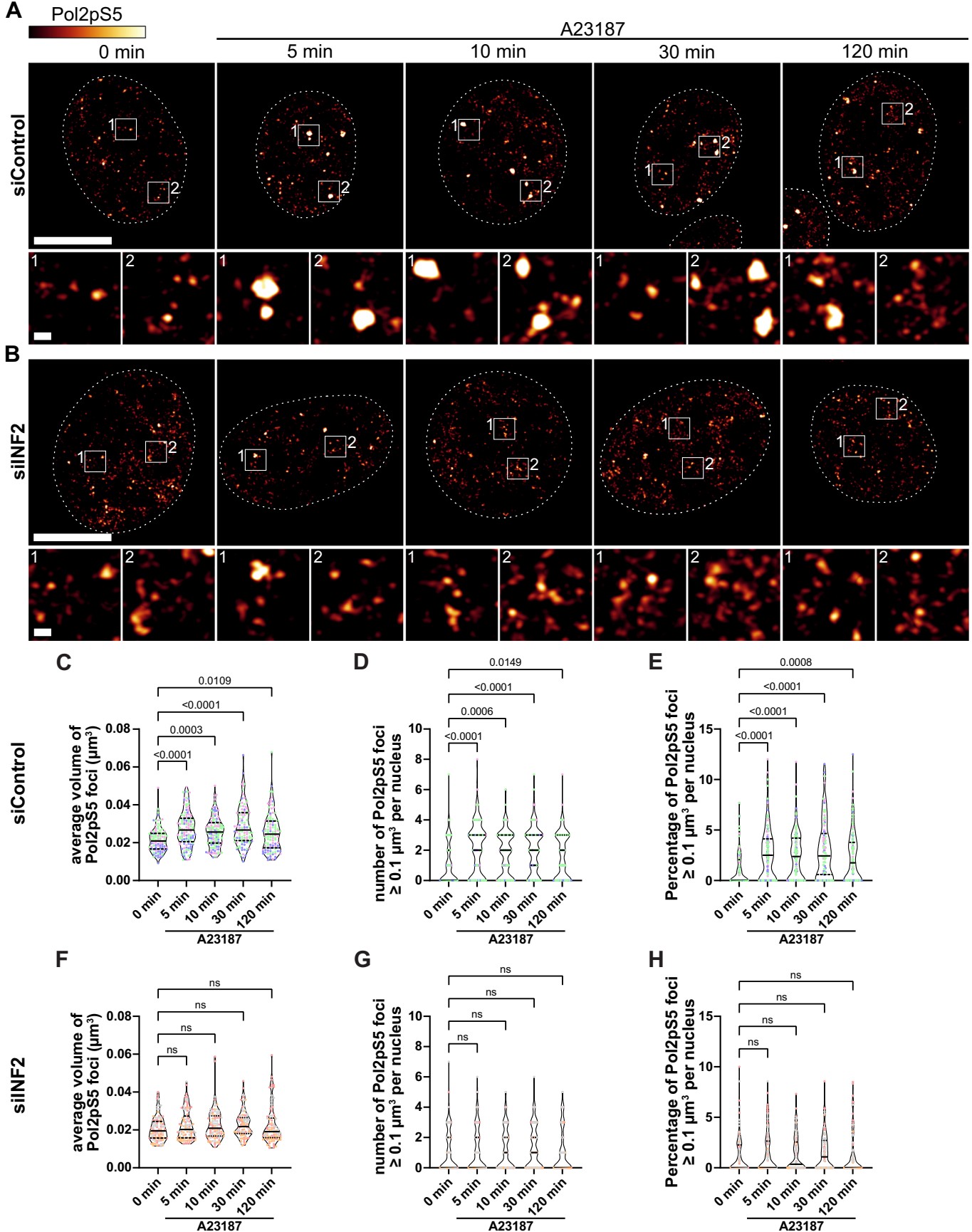

**Figure 5. RNA Pol II clustering is dependent on the actin nucleator INF2.**

(A, B) Representative images of IF staining of Pol2pS5 in NIH3T3 control cells (A) or cells depleted for INF2 (B) and stimulated with 1 µM A23187 for 0, 5, 10, 30, or 120 min prior to fixation. Nuclei are indicated by dashed lines. Gray values of detected fluorescence intensity are color-coded as depicted. White boxes indicated the magnified areas shown below. Scale bar = 10 µm (overview) or 500 nm (zoom). (C–E) Quantification of (C) average volume of detected Pol2pS5 foci (minimal volume of 0.005 µm³ used as cutoff), (D) number of Pol2pS5 foci larger than 0.1 µm³, and (E) percentage of Pol2pS5 foci larger than 0.1 µm³ from $n$ = 103, 122, 133, 101, or 113 (0, 5, 10, 30, and 120 min., respectively) siCtrl-treated cells from three independent experiments. Data are shown as violin plots with individual data points, median and quartiles. $P$ values were calculated by one-way ANOVA with Kruskal–Wallis multiple comparison test. Average volume $P$ (0 min vs. 5 min) <0.0001, $P$ (0 min vs. 10 min) = 0.0003, $P$ (0 min vs. 30 min) <0.0001, $P$ (0 min vs. 120 min) = 0.0109, number of Pol2pS5 foci $P$ (0 min vs. 5 min) <0.0001, $P$ (0 min vs. 10 min) = 0.0006, $P$ (0 min vs. 30 min) <0.0001, $P$ (0 min vs. 120 min) = 0.0149, Percentage of Pol2pS5 foci $P$ (0 min vs. 5 min) <0.0001, $P$ (0 min vs. 10 min) <0.0001, $P$ (0 min vs. 30 min) <0.0001, $P$ (0 min vs. 120 min) = 0.0008. (F–H) Quantification of (F) average volume of detected Pol2pS5 foci (minimal volume of 0.005 µm³ used as cutoff), (G) number of Pol2pS5 foci larger than 0.1 µm³, and (H) percentage of Pol2pS5 foci larger than 0.1 µm³ from $n$ = 94, 96, 91, 90, or 89 (0, 5, 10, 30, and 120 min., respectively) siINF2-treated cells from three independent experiments. Data are shown as violin plots with individual data points, median, and quartiles. Statistical analysis was performed by one-way ANOVA with the Kruskal–Wallis multiple comparison test. n.s. not significant. Source data are available online for this figure.

suspension left on ice for 30 min. Lysed cells were shortly sonicated (3×1 s, 25% power) and centrifuged for 15 min at 16,000× $g$. A sample of the supernatant was collected as "input" and the remaining solutions were either mixed or separately incubated with previously washed Anti-Flag M2 beads (Merck) by tumbling for 150 min at 4 °C. Immunoprecipitates were centrifuged and washed four times with lysis buffer (1000× $g$, 3 min). Bound protein was eluted from the beads by boiling in 60 µl of 1× Laemmli sample buffer (50 mM Tris pH 6.8, 2% SDS, 100 mM DTT, 10% glycerol, 0.01% bromophenol blue) containing 5% β-mercaptoethanol for 10 min. The protein-containing supernatant was analyzed by immunoblotting.

## Immunoblotting

Independent of prior treatment, cells were lysed directly in 1× Laemmli sample buffer and heated at 95 °C for 10 min. An equal volume was then applied for subsequent separation by sodium dodecyl sulfate-polyacrylamide gel electrophoresis (SDS-PAGE) on separating gels with varying concentrations of 8%, 10% or 12% depending on the protein size. Following semi-dry transfer (Power Blotter XL System, Invitrogen), PVDF membranes were blocked in 5% milk for 1 h at room temperature (RT) and incubated with primary antibodies overnight at 4 °C. Horseradish peroxidase-labeled secondary antibody was applied for 1 h at RT, and protein detection was carried out using enhanced chemiluminescence (ECL) reagent (SuperSignal West Femto Maximum Sensitivity Substrate, Thermo Fisher Scientific). Primary antibodies used were: mAb to α-tubulin (1:1000, CST, no. 2125), mAb to emerin (1:500, CST, no. 30853), pAb to INF2 (1:500, Proteintech, no. 20466-1-AP), mAb to SUN2 (1:1000, Abcam, no. ab124916), mAb to Flag (DYKDDDDK) tag (1:1000, CST, no. 14793), mAb to Syne3 (1:1000, OriGene, no. AM33013PU-N), pAb to Syne2 (1:1000, Invitrogen, no. PA5-78438), mAb to Syne1 (1:1000, Abcam, ab192234). Secondary antibody used was anti-rabbit IgG-HRP (1:3000, CST, no. 7074).

## Live-cell spinning disc confocal microscopy and quantification

For knockdown experiments, NIH3T3 cells stably expressing the GFP-tagged nuclear actin chromobody were initially cultured on six-well plates and transfected with the corresponding siRNAs, as detailed previously. Cells were re-plated onto Ibidi eight-well chamber slides

one day prior to imaging. On the following morning, the cells were replenished with fresh media prior to conducting live imaging at 37 °C, in a 5% $CO_2$ chamber. Images were acquired with a Yokogawa CSU-X1 spinning disc using a 100×/1.46 oil objective and Photometrics Prime BSI camera and drugs (A23187 or thrombin) were added to the cells at the microscope while scanning. Time-lapse images were recorded for up to 4 min, and the cells displaying nuclear F-actin assembly were scored as positive. Conversely, cells lacking nuclear actin structures were classified as negative. To quantify these events, the number of positive occurrences was divided by the total cell count within the field of view, expressed as a percentage of cells. Data from all wells in a chamber slide imaged on the same day were combined and presented as a single data point. Experiments for each treatment group were repeated three to four times.

## Proximity ligation assay (PLA) and analysis

The PLA was performed using the Duolink In Situ labeling kit reagents according to the manufacturer´s protocol (Sigma-Aldrich). Briefly, NIH3T3 cells were seeded onto 12-well plates and transfected with 10 pmol siSUN2 or siCtrl. After 48 h, cells were reseeded onto fibronectin-coated coverslips. The following day, cells underwent careful washing and were either left untreated, pre-treated with 10 µM BAPTA-AM for 30 min and/or treated with 1 µM A23187 for 1 min before pre-extraction in CSK buffer (10 mM PIPES pH 6.8, 100 mM NaCl, 300 mM sucrose, 3 mM magnesium chloride, 1 mM EGTA, and 0.5% Triton X-100) for 10 min at RT. Cells were then fixed with 70% ice-cold ethanol diluted in CSK buffer for 10 min at RT, and nonspecific binding was blocked with Duolink Blocking solution at 37 °C for 1 h. Primary antibodies were diluted in Duolink Antibody Diluent and incubated overnight at 4 °C in a humid chamber. Coverslips were then washed with buffer A (0.01 M Tris, 0.15 M NaCl and 0.05% Tween 20) and incubated with secondary PLA probes (anti-mouse minus and anti-rabbit plus probe) diluted in Duolink Antibody Diluent for 1 h at 37 °C in a humid chamber. Control experiments were conducted by omitting both primary antibodies or each primary antibody individually. Following another wash with buffer A, the ligation mixture was freshly prepared by diluting 5× Ligation buffer and 40× ligase in high-purity water, and applied onto coverslips which were incubated in a humid chamber for 30 min at 37 °C. After washing with buffer A, cells were incubated for 100 min at 37 °C with the amplification mix that was freshly prepared from diluting 5× Amplification

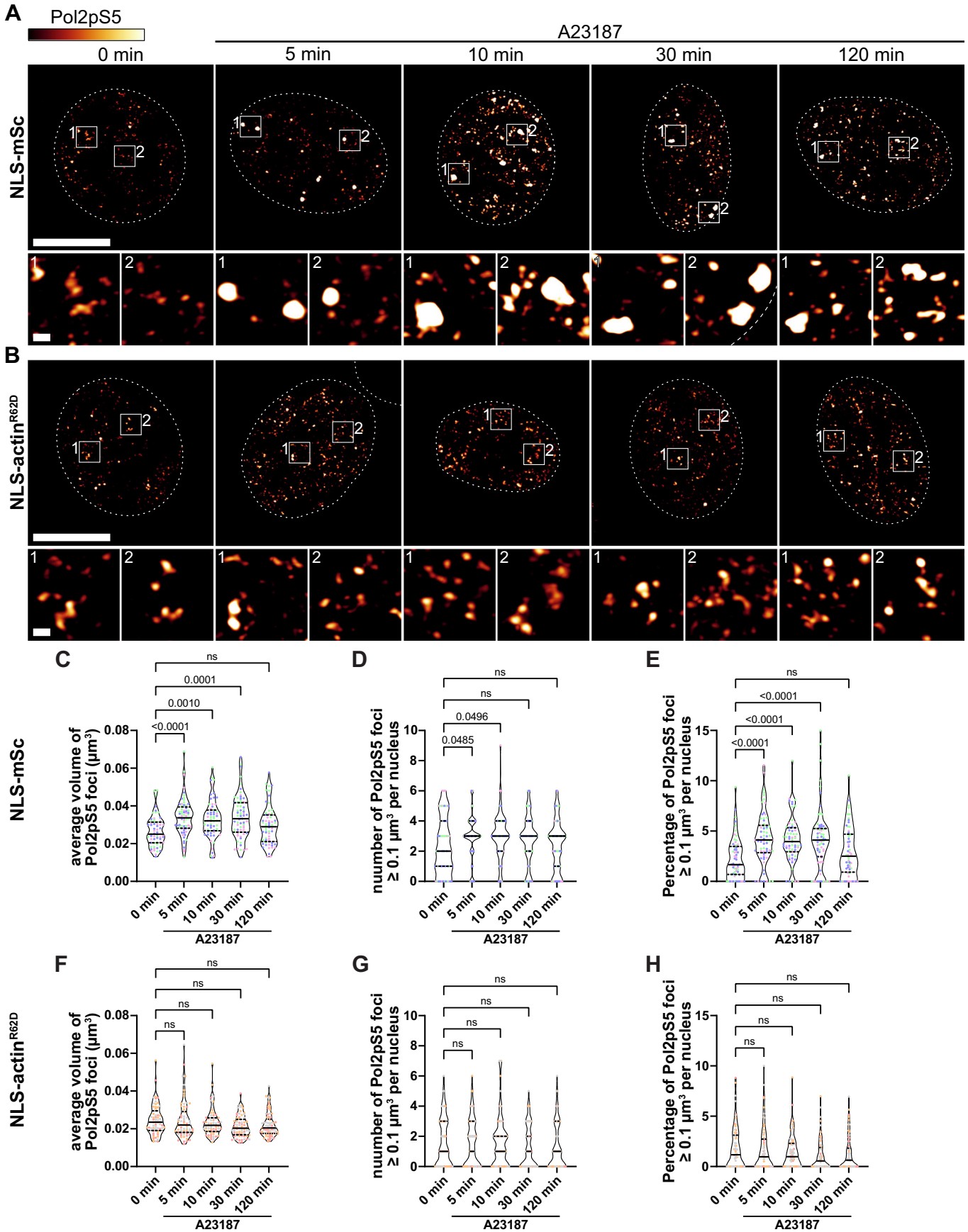

**Figure 6.   RNA Pol II clustering is dependent on dynamic nuclear actin assembly.**

(A, B) Representative images of IF staining of endogenous Pol2pS5 in NIH3T3 control cells transiently transfected with myc-NLS-mScarlet (NLS-mSc) (A) or myc-NLS-mScarlet-Actin$^{R62D}$ (NLS-actin$^{R62D}$) (B) and stimulated with 1 µM A23187 for 0, 5, 10, 30, or 120 min prior to fixation. Nuclei are indicated by dashed lines. Gray values of detected fluorescence intensity are color-coded as depicted. White boxes indicated the magnified areas shown below. Scale bar = 10 µm (overview) or 500 nm (zoom). (C–E) Quantification of (C) average volume of detected Pol2pS5 foci (minimal volume of 0.005 µm$^3$ used as cutoff), (D) number of Pol2pS5 foci larger than 0.1 µm$^3$, and (E) percentage of Pol2pS5 foci larger than 0.1 µm$^3$ from $n$ = 66, 68, 65, 69 or 75 (0, 5, 10, 30, and 120 min., respectively) cells from three independent experiments. Average volume $P$ (0 min vs. 5 min) <0.0001, $P$ (0 min vs. 10 min) = 0.0010, $P$ (0 min vs. 30 min) = 0.0001, number of Pol2pS5 foci $P$ (0 min vs. 5 min) = 0.0485, $P$ (0 min vs. 10 min) = 0.0496, Percentage of Pol2pS5 foci $P$ (0 min vs. 5 min) <0.0001, $P$ (0 min vs. 10 min) <0.0001, $P$ (0 min vs. 30 min) <0.0001. (F–H) Quantification of (F) average volume of detected Pol2pS5 foci (minimal volume of 0.005 µm$^3$ used as cutoff), (G) number of Pol2pS5 foci larger than 0.1 µm$^3$, and (H) percentage of Pol2pS5 foci larger than 0.1 µm$^3$ from $n$ = 61, 65, 63, 67 or 60 (0, 5, 10, 30, and 120 min., respectively) cells from three independent experiments. Data are shown as violin plots with individual data points, median and quartiles. $P$ values were calculated by one-way ANOVA with Kruskal–Wallis multiple comparison test. ns not significant. Source data are available online for this figure.

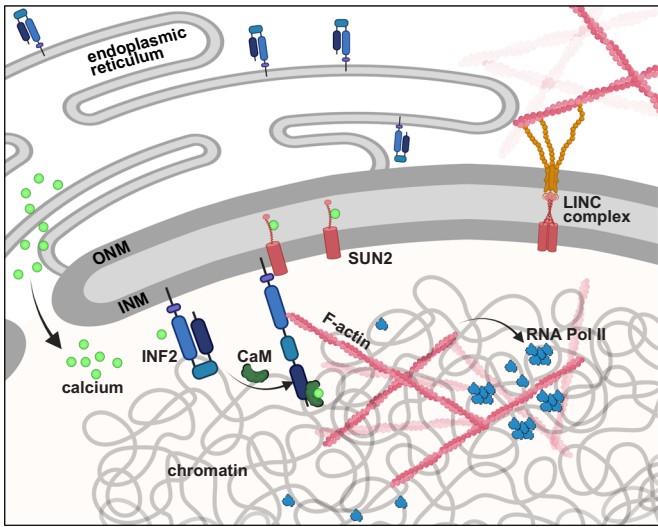

**Figure 7.   Working model.**

Illustration of the working model showing the INF2 and SUN2-mediated nuclear actin assembly leading to downstream RNA Pol II clustering. Signal-mediated influx of calcium ions to both the cytosol and the nucleus leads to activation of formin INF2 at the INM via calmodulin binding. The (monomeric form of) SUN2 potentially stabilizes the open conformation of INF2 thus allowing for rapid and transient actin filament assembly in the nuclear compartment for RNA Pol II clustering. CaM calmodulin, INF2 inverted formin 2, INM inner nuclear membrane, ONM outer nuclear membrane.

buffer and 80× polymerase in high-purity water. The cells were washed twice with buffer B (0.2 M Tris and 0.1 M NaCl), 10 min each, and DAPI was added to the second wash step. Finally, cells were washed in 0.01× buffer B for 1 min before mounting the coverslips with ProLong Diamond Antifade Mountant (Invitrogen).

The following primary antibodies were used for PLA: pAb to INF2 (1:200, rabbit, Proteintech, 20466-1-AP); pAb to SUN2 (1:200, mouse, NovusBio, NBP2-59942); pAb to Lamin A/C (1: 400, rabbit, Proteintech, 10298-1-AP), mAb to emerin (1:500, rabbit, CST, 30853), mAb to α-tubulin (1:100, rabbit, CST, 2125).

Images were acquired on a LSM800 confocal laser-scanning microscope (Zeiss) equipped with a 63×/1.4 NA oil objective, DAPI and Cy5 filter cubes as well as the Zen blue software (Zeiss). Z-stacks with 20 individual z-slices (0.21 µm z-distance) were captured for each sample, and subsequent analysis was performed

using ImageJ/FIJI. First, maximum-intensity projections were prepared for both the PLA and DAPI channels individually. For the PLA signal, background subtraction was performed (rolling ball radius 50 px) and a binary image generated. Then, a watershed transformation was performed to separate contiguous objects in the binary mask. A binary image of the corresponding DAPI channel was generated and used as a mask to analyze the number of particles/PLA dots (size: 0.1–10) within the nuclear compartment as well as the surrounding cytosolic PLA dots. Composite images containing the DAPI and PLA projected channels were generated for visualization purposes.

## Immunofluorescence

### SUN2 and INF2 co-staining

NIH3T3 cells grown on fibronectin-coated coverslips were carefully washed with PBS and fixed with ice-cold 70% ethanol for 10 min at RT. The fixative was rinsed with PBS and cells were blocked for 1 h at RT in 5% normal goat serum (Thermo Fisher Scientific) before applying primary antibodies diluted in 1% BSA and 0.05% Triton X-100 in 1×PBS for overnight incubation at 4 °C in a humid chamber. Following three washes with PBST (0.1% Tween 20 in PBS), fluorochrome-conjugated secondary antibodies diluted in 1% BSA and 0.05% Triton X-100 in 1×PBS were added for 2 h at RT in a dark humid chamber. Cells were then rinsed with PBST and distilled water before mounting the coverslips onto glass slides using ProLong Diamond Antifade Mountant (Invitrogen). Structured illumination microscopy (SIM) imaging was performed with an ELYRA 7 microscope (Zeiss) equipped with a 63× 1.4 Oil DIC objective and a Pecon incubation chamber providing a stable temperature for all samples (Lorenzen et al, 2023; Frank et al, 2023). The acquired images were SIM processed with Zen 3.0 black edition (Zeiss) using the automated Wiener filter strength given by the "weak" end criterion of the manufacturer and then analyzed with Imaris 10.1.0. To quantify INF2 fluorescence intensity, cells were imaged as a z-stack with 0.11 µm intervals. A 3D mask of the SUN2 signal was created based on seven consecutive z-slices that exhibited its typical ring-like structure. The mean fluorescence intensity (A.U.) of INF2 was then measured within this SUN2 mask. Primary antibodies used were: pAb to INF2 (1:100, rabbit, Proteintech, 20466-1-AP); pAb to SUN2 (1:200, mouse, NovusBio, NBP2-59942); mAb to Lamin A/C (1:100, mouse, CST, 4777). Secondary antibodies used were: Alexa Fluor 647 goat anti-rabbit IgG (H + L) (1:400, Invitrogen, A21244); Alexa Fluor 488 chicken anti-mouse IgG (H + L) (1:400, Invitrogen, A21200).

### RNAPol2pS5 foci

NIH3T3 cells grown in six-well plates were either transfected with siRNA for knockdown experiments or plasmid DNA to overexpress a nuclear-localized Actin$^{R62D}$ mutant as described above. The cells were reseeded onto fibronectin-coated coverslips the next day. After an additional 24 h (for overexpression experiments) or 48 h (for knockdown experiments), the cells were treated with 1.5 μM A23187 for the indicated time, followed by fixation with 4% paraformaldehyde (PFA) for 15 min at RT. The fixative was washed out with PBS, and cells were permeabilized with 0.3% Triton X-100 in PBS for 15 min. After blocking with 1% BSA and 5% normal goat serum in PBS for 60 min at RT, primary antibodies diluted in 1% BSA and 0.05% Triton X-100 in PBS were applied to the cells and incubated for 2 h at RT in a humid chamber. Following three washes with PBST, fluorochrome-conjugated secondary antibodies and DAPI (1:1000) diluted in 1% BSA and 0.05% Triton X-100 in 1×PBS were added for 1.5 h at RT in a dark humid chamber. Cells were washed three times with PBST and distilled water before mounting the coverslips onto glass slides using ProLong Diamond Antifade Mountant (Invitrogen).

Primary antibodies used were: mAb to RNA Pol II CTD S5ph (1:200, rat, Active Motif, 61701); mAb to SUN2 (1:400, rabbit, Abcam, ab124916). Secondary antibodies used were: Alexa Fluor 647 goat anti-rabbit IgG (H + L) (1:1000, Invitrogen, A21244); Alexa Fluor 488 goat anti-rat IgG (H + L) (1:1000, Invitrogen, A11006).

Fluorescence images were generated using a LSM800 confocal laser-scanning microscope (Zeiss) equipped with a 63×, 1.4 NA oil objective and Airyscan detector, along with the Zen blue software (Zeiss). Cells were imaged as a z-stack of 22 z-slices with a 0.18 μm interval. All images acquired for one independent experiment were subjected to airyscan processing with Zen blue using consistent Super Resolution parameters. 3D rendering of RNAPol2pS5 foci was performed based on fluorescence intensities of the RNAPol2 channel using the Surface Tool of Imaris 10.1.0. For the quantification of average RNAPol2pS5 foci volume per nucleus, we excluded foci smaller than 0.005 μm³ due to limitation in z-resolution.

### Nesprin staining

NIH3T3 cells grown in 12-well plates were transfected with either pEF-mScarlet-dnKASH or pEF-mScarlet DNA as previously described. The following day, the cells were reseeded onto fibronectin-coated coverslips and, 24 h later, fixed in 4% methanol-free PFA for 15 min at RT. The fixative was washed out with PBS and cells were permeabilized with 0.3% Triton X-100 in PBS for 15 min. After blocking with 1% BSA and 5% normal goat serum in PBS for 60 min at RT, primary antibodies diluted in 1% BSA and 0.05% Triton X-100 in PBS were applied to the cells and incubated overnight at 4 °C in a humidified chamber. Following three washes with PBST, fluorochrome-conjugated secondary antibodies and DAPI (1:1000) diluted in 1% BSA and 0.05% Triton X-100 in 1×PBS were added for 1.5 h at RT in a dark humid chamber. Cells were washed three times with PBST and distilled water before mounting the coverslips onto glass slides using ProLong Diamond Antifade Mountant (Invitrogen).

Primary antibodies used were: pAb to Syne2 (1:200, rabbit, Invitrogen, PA5-78438); mAb to Syne3 (1:200, mouse, OriGene, AM33013PU-N).

Secondary antibodies used were: Alexa Fluor 647 goat anti-rabbit IgG (H + L) (1:400, Invitrogen, A21244); Alexa Fluor 488 goat anti-mouse IgG (H + L) (1:400, Invitrogen, A11029).

### Statistical analysis

Statistical analyses were performed using Graph Pad 10.1.0 software. Data are presented either as column bar graph ± SD or truncated violin plots showing all data points with median and quartiles. For comparison of normally distributed data from two groups, a two-tailed $t$ test was performed. For the statistical analysis of more than two groups, one-way analysis of variance (ANOVA) with the corresponding multiple comparison test was applied as indicated in the figure legends. Numeric p values are indicated in the figures. For all analyses shown, at least three independent experiments were performed.

### Graphics

BioRender was used to prepare Figs. 1B and 7 and the synopsis image of this study.

## Data availability

This study includes no data deposited in external repositories.

The source data of this paper are collected in the following database record: biostudies:S-SCDT-10_1038-S44319-024-00274-8.

## Peer review information

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

## Acknowledgements

The authors thank our laboratory members for their helpful discussions. The authors thank Antje Müller and Peter Gebhardt for technical assistance. pEGFP-N3-hSUN2 was a gift from Ulrike Kutay. The authors thank our funders for their generous support. Germany's Excellence Strategy; EXC-2189, project ID 390939984 (RG). German Research Foundation (DFG) grant GR 2111/13-1 (RG).

## Author contributions

**Svenja Ulferts**: Formal analysis; Investigation; Visualization; Writing—original draft; Writing—review and editing. **Robert Grosse**: Conceptualization; Supervision; Funding acquisition; Writing—original draft; Project administration; Writing—review and editing.

Source data underlying figure panels in this paper may have individual authorship assigned. Where available, figure panel/source data authorship is listed in the following database record: biostudies:S-SCDT-10_1038-S44319-024-00274-8.

## Funding

## Disclosure and competing interests statement

The authors declare no competing interests.

# Expanded View Figures

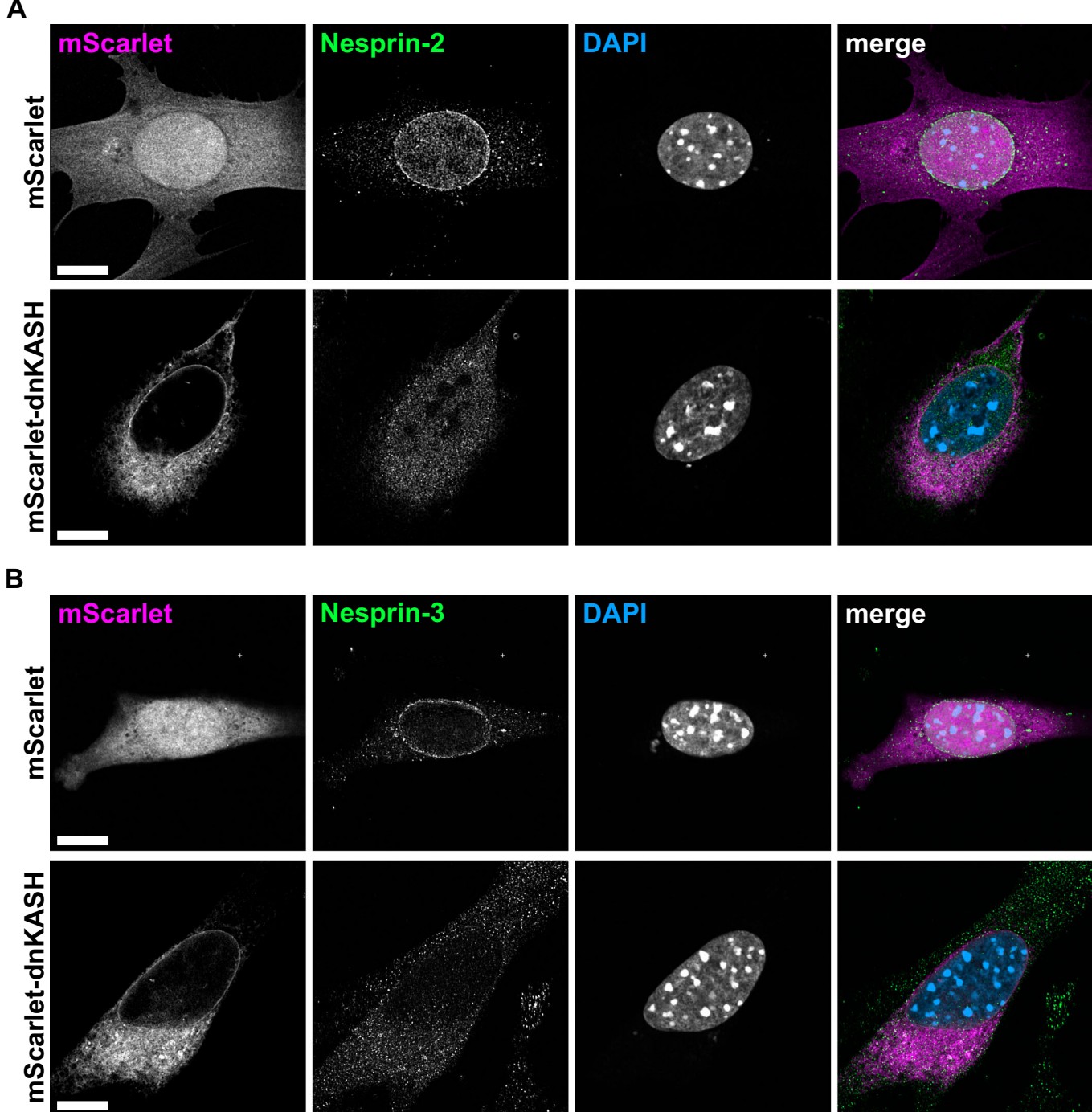

**Figure EV1.  Dominant-negative nesprin construct displaces endogenous nesprins from nuclear envelope.**

(A, B) Immunofluorescence images of NIH3T3 cells transiently expressing mScarlet (magenta) or mScarlet-dnKASH (magenta). Cells were stained for (A) nesprin2 (green) or (B) nesprin3 (green) and DNA (DAPI, blue). scale bar = 10 μm.

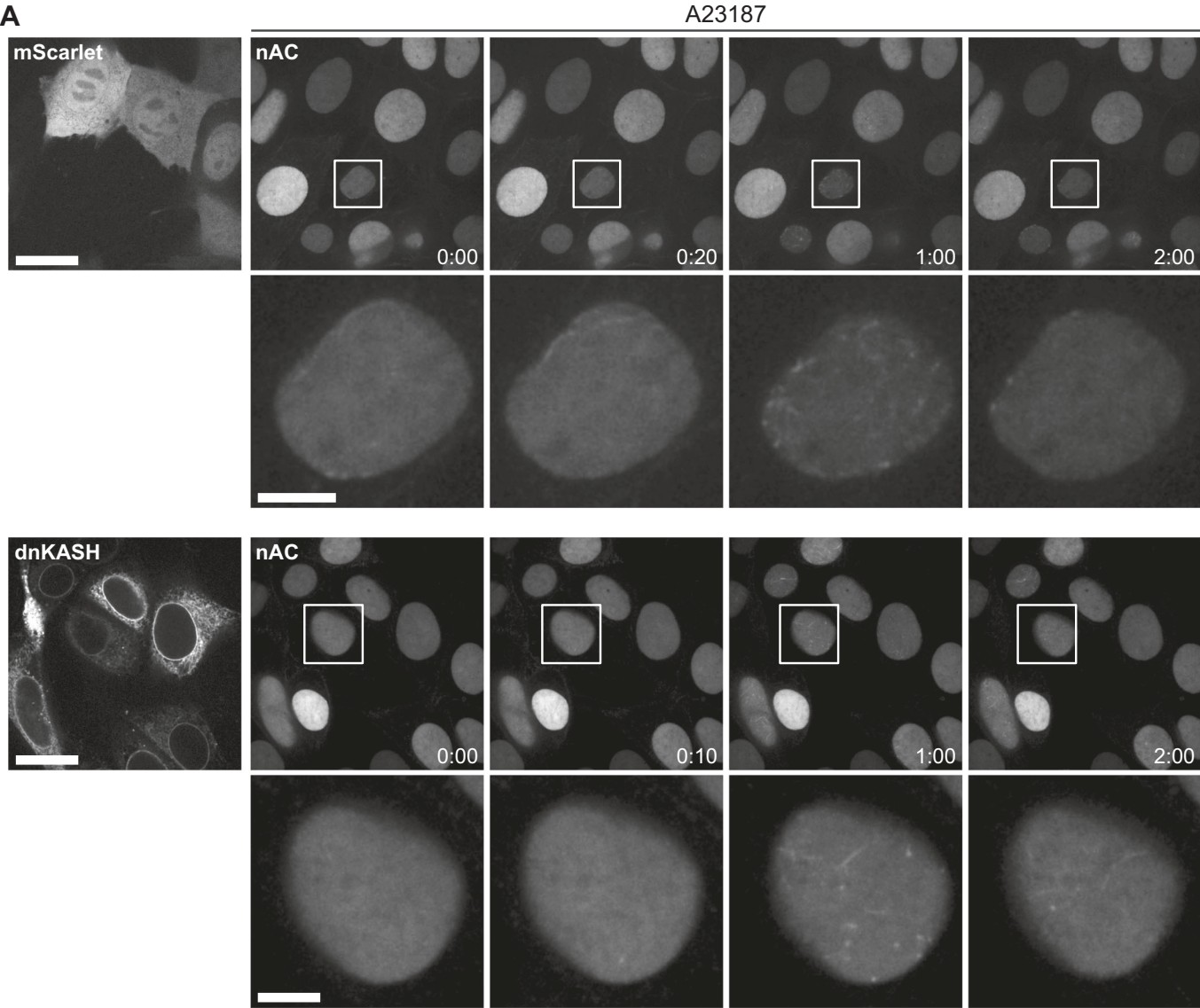

**Figure EV2. Calcium transients trigger nuclear actin assembly independent of the mechanotransduction function of SUN2.**

(A) Representative spinning disc confocal slices of NIH3T3 fibroblasts stably expressing nAC-tagGFP. Cells were transfected with mScarlet-tagged dominant-negative KASH (dnKASH) or mScarlet empty vector and stimulated with 1 μM A23187 to induce rapid nuclear actin assembly. Cells were quantified for positive events as analyzed in Fig. 1F. White boxes indicate magnified areas shown below the overview images. Scale bar = 20 μm (overview) or 5 μm (zoom). (min:s after drug treatment).

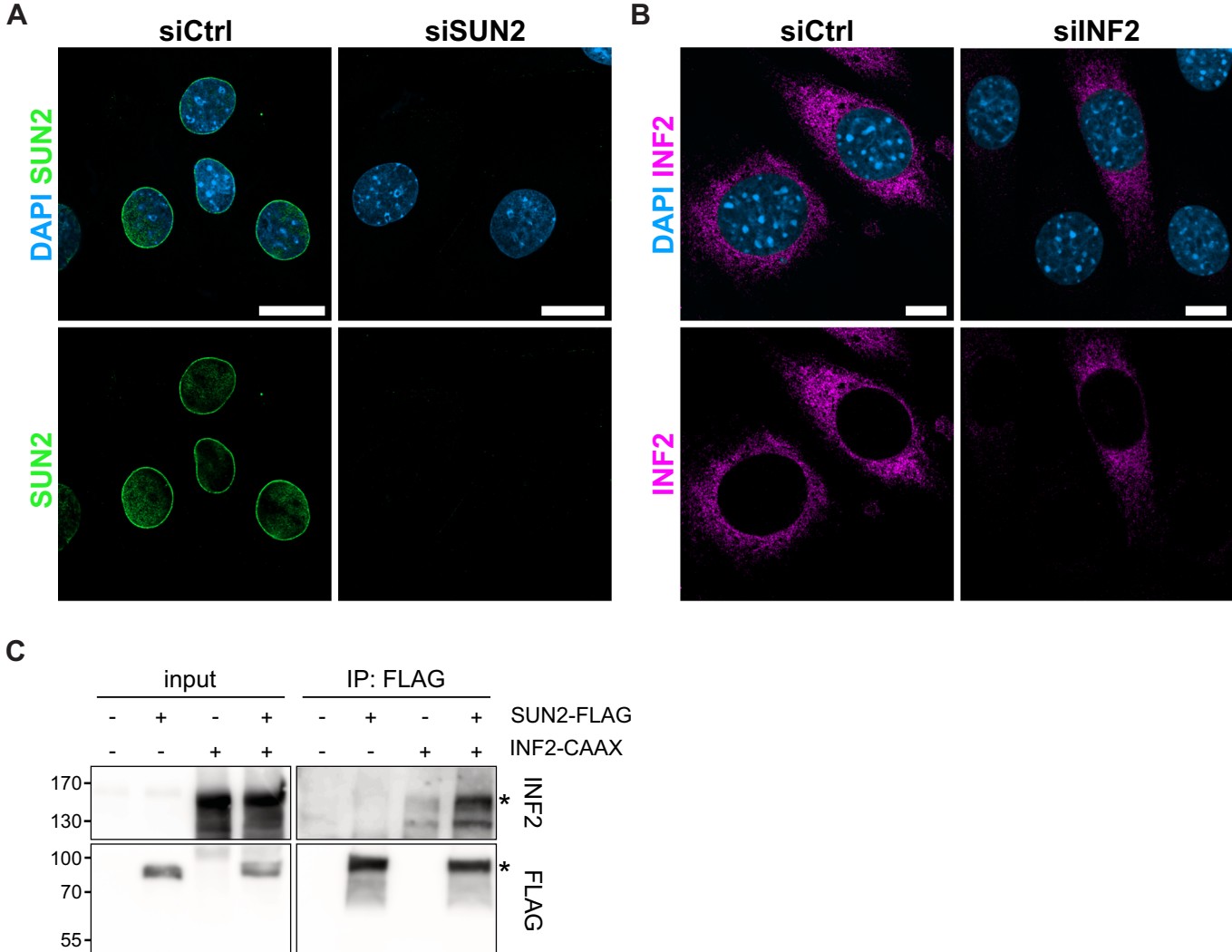

**Figure EV3. Immunostaining showing antibody specificity and Co-IP of SUN2 and INF2.**

(A) Immunostaining of endogenous SUN2 in NIH3T3 cells reveals antibody specificity as well as efficient knockdown of SUN2 protein upon transfection with SUN2 targeting siRNA. Scale bar = 20 μm. (B) Immunostaining of endogenous INF2 in NIH3T3 cells reveals antibody specificity as well as efficient knockdown of INF2 protein upon transfection with INF2 targeting siRNA. Scale bar = 10 μm. (C) Co-immunoprecipitation of HEK293T cells expressing SUN2-FLAG and INF2-CAAX using anti-FLAG beads. Asterisks indicate co-immunoprecipitated INF2 (upper right panel) and SUN2-FLAG (lower right panel). Associated INF2 was detected using an anti-INF2 antibody.

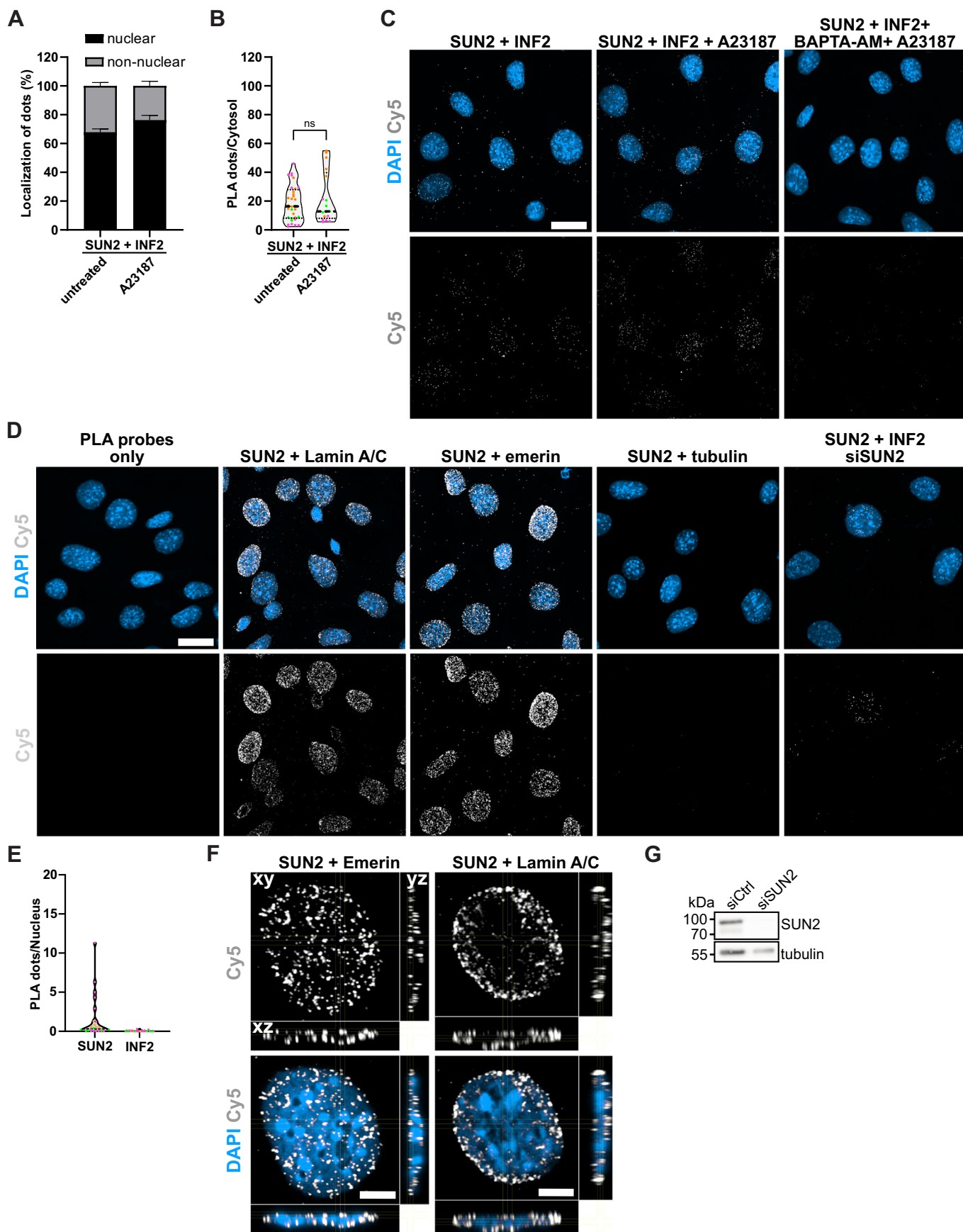

◀  **Figure EV4.  PLA control experiments show specificity of applied technique.**

(A) Cellular distribution of detected PLA signals. Each bar represents mean $+/-$ SEM from at least three independent experiments. (B) Quantification of cytosolic PLA signal reveals no significant increase in number upon A2387 treatment. Data are shown as violin plot with median and quartiles. Individual data points represent one field of view acquired from three independent biological replicates. two-tailed *t* test. ns, not significant. (C) Representative MIPs of NIH3T3 cells subjected to PLA and quantified in Fig. 3D. Discrete PLA signals (Cy5) indicate the interaction of endogenous target proteins. Scale bar $= 20\,\mu m$. (D) Proximity ligation assay (PLA) of NIH3T3 cells using the indicated primary antibodies. PLA probes alone serve as technical control to detect nonspecific binding of PLA probes. Use of SUN2 primary antibody together with antibodies directed against the known interactors Lamin A/C or emerin serve as positive control. Primary antibody against cytoplasmic tubulin serves as negative biological control. Images show representative maximum-intensity projections (MIP) of 20 confocal z-slices (0.21 μm z-distance). Discrete PLA signals (Cy5) indicate the interaction of endogenous target proteins within the nucleus (DAPI). Scale bar $= 20\,\mu m$. (E) Quantification of PLA signal (dots/nucleus) from technical controls using indicated primary antibodies alone. Data are shown as violin plot with median and quartiles. Individual data points represent one field of view acquired from three independent biological replicates. (F) Orthogonal optical cross section reconstructed from a Z-scan through a NIH3T3 cell nucleus stained for DAPI and PLA dots (Cy5) between SUN2 and emerin or SUN2 and Lamin A/C. Scale bar $= 5\,\mu m$. (G) Immunoblot showing siRNA-mediated SUN2 knockdown efficiency of cells subjected to PLA (Fig. 3A,B).

