## [Peer Review File · EMBO Reports]

SUN2 mediates calcium-triggered nuclear actin polymerization to cluster active RNA polymerase II

Svenja Ulferts and Robert Grosse

Corresponding author(s): Robert Grosse (robert.grosse@pharmakol.uni-freiburg.de), Svenja Ulferts (svenja.ulferts@pharmakol.uni-freiburg.de)

Review Timeline:

Submission Date:	23rd Feb 24
Editorial Decision:	3rd Apr 24
Revision Received:	30th Jul 24
Editorial Decision:	23rd Aug 24
Revision Received:	10th Sep 24
Accepted:	15th Sep 24

Editor: Deniz Senyilmaz Tiebe

Transaction Report:

Dear Robert,

Thank you for the submission of your research manuscript to our journal, which was now seen by three referees, whose reports are copied below.

My apologies for this unusual delay in getting back to you. It took longer than anticipated to receive the full set of referee reports.

Referees express interest in the proposed LINC independent role of SUN2 in calcium induced nuclear F-actin assembly and RNAP II clustering. However, they also raise significant concerns that need to be addressed to consider publication here.

I believe that the reports are constructive and performing the proposed experiments will strengthen the manuscript. Given these positive recommendations, we would like to invite you to submit a revised manuscript. Please revise your manuscript with the understanding that the referee concerns (as in their reports) must be fully addressed and their suggestions taken on board. Please address all referee concerns in a complete point-by-point response. Acceptance of the manuscript will depend on a positive outcome of a second round of review. It is EMBO reports policy to allow a single round of major experimental revision only and acceptance or rejection of the manuscript will therefore depend on the completeness of your responses included in the next, final version of the manuscript.

We realize that it is difficult to revise to a specific deadline. In the interest of protecting the conceptual advance provided by the work, we recommend a revision within 3 months. Please discuss the revision progress ahead of this time with me if you require more time to complete the revisions, or if you have questions or comments regarding the revision (also by video chat).

1. A data availability section providing access to data deposited in public databases is missing (where applicable).
2. Your manuscript contains statistics and error bars based on $n=2$. Please use scatter plots in these cases.

You can submit the revision either as a Scientific Report or as a Research Article. For Scientific Reports, the revised manuscript can contain up to 5 main figures and 5 Expanded View figures, and it should not exceed 27000 characters. If the revision leads to a manuscript with more than 5 main figures it will be published as a Research Article. In this case the Results and Discussion section should be separate. If a Scientific Report is submitted, these sections have to be combined. This will help to shorten the manuscript text by eliminating some redundancy that is inevitable when discussing the same experiments twice. In either case, all materials and methods should be included in the main manuscript file.

4) a .docx formatted letter INCLUDING the reviewers' reports and your detailed point-by-point responses to their comments. As part of the EMBO publication's Transparent Editorial Process, EMBO reports publishes online a Review Process File (RPF) to accompany accepted manuscripts. This File will be published in conjunction with your paper and will include the referee reports, your point-by-point response and all pertinent correspondence relating to the manuscript.

<https://www.embopress.org/page/journal/14693178/authorguide#transparentprocess>

5) a complete author checklist, which you can download from our author guidelines

<https://www.embopress.org/page/journal/14693178/authorguide>. Please insert information in the checklist that is also reflected in the manuscript. The completed author checklist will also be part of the RPF.

6) Please note that all corresponding authors are required to supply an ORCID ID for their name upon submission of a revised manuscript (<<https://orcid.org/>>). Please find instructions on how to link your ORCID ID to your account in our manuscript tracking system in our Author guidelines

<<https://www.embopress.org/page/journal/14693178/authorguide#authorshipguidelines>>

7) Before submitting your revision, primary datasets produced in this study need to be deposited in an appropriate public database (see <https://www.embopress.org/page/journal/14693178/authorguide#datadeposition>). Please remember to provide a reviewer password if the datasets are not yet public. The accession numbers and database should be listed in a formal "Data Availability" section placed after Materials & Method (see also

<https://www.embopress.org/page/journal/14693178/authorguide#datadeposition>). Please note that the Data Availability Section is restricted to new primary data that are part of this study. * Note - All links should resolve to a page where the data can be accessed. *

Additional information on source data and instruction on how to label the files are available:

<https://www.embopress.org/page/journal/14693178/authorguide#sourcedata>

9) Our journal encourages inclusion of *data citations in the reference list* to directly cite datasets that were re-used and obtained from public databases. Data citations in the article text are distinct from normal bibliographical citations and should directly link to the database records from which the data can be accessed. In the main text, data citations are formatted as follows: "Data ref: Smith et al, 2001" or "Data ref: NCBI Sequence Read Archive PRJNA342805, 2017". In the Reference list, data citations must be labeled with "[DATASET]". A data reference must provide the database name, accession number/identifiers and a resolvable link to the landing page from which the data can be accessed at the end of the reference. Further instructions are available at <http://www.embopress.org/page/journal/14693178/authorguide#referencesformat>

10) Regarding data quantification (see Figure Legends:

<https://www.embopress.org/page/journal/14693178/authorguide#figureformat>)

- the name of the statistical test used to generate error bars and P values,
- the number (n) of independent experiments (please specify technical or biological replicates) underlying each data point,
- the nature of the bars and error bars (s.d., s.e.m.),
- If the data are obtained from n Program fragment delivered error ``Can't locate object method "less" via package "than" (perhaps you forgot to load "than"?) at //ejpvfs23/sites23b/embo/www/letters/embo_decision_revise_and_review.txt line 56.' 2, use scatter blots showing the individual data points.

12) Please also note our reference format:

I look forward to seeing a revised version of your manuscript when it is ready. Please let me know if you have questions or comments regarding the revision.

Kind regards,

Deniz

Deniz Senyilmaz Tiebe, PhD
Scientific Editor
EMBO Reports

Referee #1:

In this manuscript, the authors Ulferts et al, are dissecting the relationship between calcium-triggered nuclear actin polymerization and the nucleoskeleton, and what the functional consequences of disrupting this are on transcription. They find that SUN2 is required for nuclear f-actin assembly, and that this function is independent of SUN2's role in the LINC complex, that SUN2 and INF2 colocalize at the inner nuclear membrane in a calcium dependent manner, that SUN2 drives RNAP II clustering, and that this clustering is dependent on nuclear actin polymerization. The manuscript and figures are generally well organized and demonstrative of their data. Listed below are suggestions to improve the clarity and address some issues with the manuscript.

Major:

1. Line 89 - "actin filaments for euchromatin formation." It is not clear what you mean by this.
2. Line 90-92: What about INF2 is associated with disease? Please be more explicit about how INF2 (i.e. mutations within the protein, abnormal expression, dysfunction) contributes to disease pathologies.
3. Line 100 - "which receptors" - it is unclear what you are referring to by this.
4. Line 102-105 - the conclusions of the paper are slightly overstated. INF2 is never tested in the clustering of RNA Pol II. Further the statement of "disease-associated formin INF2" makes it seem you are testing a disease mutation and its role in this process.
5. Line 109-113. The rationale behind the experiments are not fully clear. In particular, the second sentence is hard to follow.
6. In the first paragraph of the Results it is important to explain the use of A23187
7. Figure 1 - it is not clear what you are measuring when you say nuclear F-actin (% of cells). A description is needed in the methods (and maybe the figure legend). It would also be clearer if the axis label was % of cells with nuclear F-actin.
8. Specificity of the INF2 antibody for immunofluorescence is not shown.
9. Figure 2 - zoom in images of single nuclei would make it easier on the reader to see the differences.
10. It would be easier to compare the data in figures 4 and 5 if in Fig 4 the images were next to each other and in both Fig. 4 and 5 the data were graphed on the same graph (ex. Control and siSUN2). Also the graph labels could be clearer - stating "number of" and "Percentage of" for current 4C/D, G/H and similarly in Fig 5.
11. In Figure 5, as a reader, I prefer to see the images in addition to the graphs of the quantification. I recommend showing just two time points.
12. It is surprising the NLS-actin experiment was done in Figure 5 and not siINF2. The rationale for this change is not clear. Also the data for Figure 5 could be better explained in the results.
13. The conclusions of the paper could be strengthened by showing actin filaments associated with the RNA Pol II clusters and testing the role of INF2 in the clustering. Further, showing additional evidence the RNA Pol II clusters are actively transcribing would strengthen the paper.
14. Lines 172-174 - conclusion is slightly overstated. INF2 is never tested in RNA Pol II clustering, nor are the actin filaments shown to directly interact with RNA Pol II.

Minor:

1. Line 78 - change "site" to "side" to illustrate where the nesprins connect to the nuclear envelope. "Site" makes it seem like there is only one specific point of connection whereas "side" makes it clearer that the connection can happen anywhere along the nuclear envelope.
2. Line 85 - add signaling after serum response factor.

3. Line 87 and elsewhere - "we could previously show" is odd phrasing. Recommend changing to "we previously showed/found"
4. Line 115 - comma needed "SUN2 like INF2" to "SUN2, like INF2"
5. Line 129 - Figure callout is wrong. (Fig 2C) should be (Fig 2B)
6. Line 133 - Figure callout is wrong. (Fig 3C) should be (Fig 2C)
7. Line 165-167 - seems like this sentence belongs in the prior paragraph and this paragraph should start with the NLS-actin.
8. Line 196 - what is "the formin" referring to? INF2?
9. Line 207 - it is unclear what you mean by "autonomously"
10. The model could be better walked through in the discussion. Also, while KASH domains are not required for the SUN2 functions in the paper, they could still be bound to the SUN2 involved. This should be stated in the model figure legend.
11. The calcium elevation using A23187 is not described in the methods
12. It is unclear if the quantifications were performed in a genotypically blinded manner.
13. Figure 1 - change labeling of westerns to be F, and make graph E.
14. Figure 1 - why are the statistics used in 1C and 1F different.
15. Figure 1B - referring to the color of the components you are talking about in the legend would help make it clearer.
16. Figure 2C - were the IPs done in high calcium or not?
17. I am surprised the full western blots are not provided as supplemental figures. It would also be nice if the raw data for the quantifications were provided as a supplemental file.

Referee #2:

This work follows up on a previous study by these authors, showing that the formin INF2 can cause actin polymerization in the nucleus in response to stimuli that elevate calcium, in addition to its known role in cytoplasmic actin polymerization. This INF2-mediated actin causes changes in chromatin organization. In the present manuscript, the authors show that the inner nuclear membrane (INM) protein SUN2 is necessary for this response to increased cytoplasmic actin, possibly through interaction with INF2. The actin polymerization is very rapid, peaking within 1 min of stimulus and dissipating within 2 min or so. The authors show that calcium stimulation also causes an increase in active clusters of RNA polymerase II, in a manner that is dependent on SUN2.

This work provides an interesting extension to the growing list of functions performed by actin in the nucleus. In addition, by tying SUN2 to the process the work links the actin to the INM. The experiments as presented are well controlled in general. The work, however, is quite 'thin', not delving into any of the results shown in any detail, and falling short on providing real insight. Additional experiments to raise significance would be useful.

Specific Comments:

- 1) In Figure 1, the authors quantify the response to A23187 treatment by % cells demonstrating nuclear actin. It would be useful to have a more detailed characterization of the response. For example, in Figure 1D, the siRNA treatment of SUN2 or INF2 results in a decreased percentage from ~65 to ~35. Is the actin response of the cells that still were classified as "positive" weaker than in siCtrl? Such an analysis would be more informative than what is given at present.
- 2) Figure 1C and Figure 1F. These are negative results for emerin and LINC effects on actin polymerization. It would be useful to have data showing that the treatments (emerin siRNA and dominant-negative KASH construct) are affecting known processes. Also, it would be useful to have some indication of the expression level of the dominant-negative KASH construct (western blot or cell staining).
- 3) It is not clear that the authors really 'prove' that SUN2 is needed for nuclear actin polymerization independently of its role as a component of LINC. While the dominant-negative KASH construct is a start, additional experimentation (for example, siRNA of nesprins) would strengthen the result.
- 4) Figure 2A and 2B show imaging of SUN2 and INF2 on the nuclear membrane, but that the two do not necessarily co-localize there, with SUN2 being quite homogenous whereas INF2 is somewhat punctate. It is, therefore, unclear what the significance of this figure is, considering that the authors' previous paper (reference 16) shows this localization for INF2, and that SUN2 is well known to localize to the INM. Given the finding that calcium causes actin polymerization dependent on both SUN2 and INF2, it would be interesting to test whether INF2 and/or SUN2 distribution changes on the same time scale as the actin polymerization. This experiment would be an important addition to the PLA assays done subsequently, to show if the entire population of INF2 or SUN2 is changing following calcium addition, or whether it just appears to be a small percentage of the population.
- 5) Figure 2C shows co-IP of SUN2 and INF2. Comments:
 - a. In the methods, the authors state that the cells were transfected with hSUN2-FLAG or INF2-CAAX separately. Does this mean that the INF2 they are detecting is transfected INF2, or endogenous INF2? If transfected, that construct does not appear to be listed.
 - b. a useful negative control would be IP of emerin, another INM protein. Immunoprecipitations of membrane proteins can reveal a variety of non-specific interactions, depending on how well the membranes were detergent extracted.
 - c. Another important negative control would be to test co-IP of SUN2-FLAG with the non-ER bound isoform of INF2 (the isoform that lacks the CAAX box). If ER binding by INF2 is necessary for the interaction, the CAAX-less isoform should not bind.
- 6) Figure 3 shows PLA assays demonstrating SUN2/INF2 interaction. There are a number of comments on this analysis:
 - a. The examples shown in panel A are processed such that the PLA signal is weak. Despite this processing, there are clearly a

number of PLA dots in the cytoplasm. The authors should be more transparent about this, by displaying brighter images, and commenting on the background cytoplasmic dots. Do these also change with A23187 treatment?

b. The authors supply some excellent negative controls, but another very useful negative control would be to use another nucleoskeleton protein (such as emerin or lamin) to show that its interactions with INF2 are minimal or do not change with calcium addition.

c. Also in Figure 3, the authors quantify the number of PLA dots, but there seems also to be a change in dot brightness. Quantification on brightness per dot would be a useful added parameter.

d. It would be useful to determine whether PLA dots are actin-dependent by treating with LatA.

7) Pertaining to the Pol2pS5 clusters shown in Figure 4:

a. The formation of large Pol2pS5 clusters in as little as 5 min post-calcium increase is fascinating. However, the change in nuclear actin takes place more rapidly, and is essentially gone by 5 min (movie 1). Is it possible for the authors to test Pol2pS5 clusters at earlier time points (such as 1 min post-stimulus)?

b. It is unclear whether the formation of large Pol2pS5 clusters is dependent on INF2. Testing the siINF2 cells for an effect on these clusters is an important control.

8) In the Introduction, the authors state that INF2 exists mostly in a CAAX-box modified isoform that localizes to the ER membrane (ref 19. Chhabra et al 2009). Is this true for every cell type? Is the CAAX-lacking isoform more abundant in certain situations?

Referee #3:

Summary:

This manuscript explores the often overlooked role of the nuclear cytoskeleton in signal transduction of pro-transcriptional cues. The authors use calcium signaling as a potent model for exploring the mechanism of signaling to produce consistent, well supported observations for their experiments. A variety of imaging techniques are applied to answer specific questions about which proteins are involved in this process, and how they change their localization in response to increased intracellular calcium levels. The results directly implicate SUN2 as a primary transducer of calcium signaling, independent of the force sensing pathway governed by KASH domain containing proteins. Based on the observation that the population of elongating RNA polymerase II increases cluster size during successful calcium signaling transduction, the authors conclude that calcium signaling is converted into a transcriptional response via the reorganization of nucleoskeleton components.

Key Findings:

The key finding described by the manuscript is that RNA polymerase clustering (inferred as transcription) is directly linked to the formation of nuclear actin filaments. I consider this a significant finding, but it could be supported by the additional suggested experiments (see below). As nuclear actin continues to be a difficult to study and enigmatic topic, such mechanistic work tying the phenomenon to a critical aspect of cell biology (transcription) is likely to be of wide interest to the community. The findings are well supported by the data, I find myself particularly convinced by the use of the dominant negative forms of actin abolishing the formation of RNA polymerase clusters. The confirmation of the RNAi knockdowns with western blots is also greatly appreciated.

Major Comments:

-The model put forth by the authors is easily interpreted and supported by strong experimental evidence. Specifically, the protein components implicated in the transduction of the actin signal are thoroughly investigated, through both the use of small molecule inhibitors and siRNA knockdown experiments. However, additional interrogation about the nature of the transcriptional response would increase the impact of the publication:

(1) I am interested in the role of actin polymerization instructing RNA polymerase clustering. Specifically, would it be possible to see if actin polymerization co-localizes with the RNA polymerase clusters that appear upon calcium signaling induction? Given the reagents on hand (RNA Pol II pSer5 antibody, actin chromobody), this should be relatively straightforward to perform as an immunostain, although there might be additional technical limitations that I have not considered.

(2) Is RNA polymerase clustering induced on genes that are expressed in a response to A23187 treatment? Combining an RNA FISH stain for a A23187 responsive gene with the RNA Pol II antibody stain would directly associate the newly formed clusters with the induced transcription. This is a much larger ask since it would be the development of a new assay, and therefore I would not consider it strictly necessary. However, I would say that explicitly stating that this reorganization of Pol II is indicative of transcription (Line 102) should be tempered if this evidence is not provided.

-As mentioned before, I am impressed by the dominant negative form of actin preventing the formation of RNA polymerase clustering. I understand that this perturbation is well known in the actin field, but if it is feasible I believe it would bolster the findings to perform the actin chromobody experiment on cells expressing this construct to demonstrate that preventing polymerization of the nuclear actin filaments as the likely mechanism of action.

Minor Comments:

-Does the actin polymerization happen throughout the nucleus, or is it restricted to the nuclear lamina? Perhaps the authors could include a XZ view of the actin chromobody fluorescence (like the projections used for the PLA experiments) to visualize this.

-For supplemental figure EV1, it is difficult to see the actin polymerization in response to the drug. If the images were enlarged like they are in Figure 1, it would help avoid confusion.

-While it is very much appreciated that the authors display their data as violin plots with individual replicates labeled as different colors, it is a bit visually distracting for the viewer. The authors may want to simplify the plotting style and show more complete plots in the supplemental materials, or simply reduce the opacity of the individual data points and emphasize the horizontal lines that denote the median and quartile values.

We thank the reviewers for their valuable time and insightful comments and we are happy about their generally positive response.

We are resubmitting our completely revised paper now containing:

1. New panel in Figure 1F and H showing that A23187-induced nuclear actin assembly is independent of endogenous nesprins supporting our dominant negative KASH (dnKASH) data in Figure 1G
2. New panel in Figure 2C showing an increase in INF2 fluorescence intensity at the nuclear membrane upon A23187 treatment of the cells
3. New Figure 5 showing the role of INF2 in A23187 induced RNA Pol II cluster formation
4. New panels in Figure 6A and B (previously Figure 5) showing the example images to complement the graphs of the RNA Pol II cluster quantification
5. New Figure EV1 showing the displacement of endogenous nesprins from the nuclear membrane upon overexpression of dnKASH
6. New panel in Figure EV3B showing specificity of the INF2 antibody in siRNA depleted cells
7. New panels in Figure EV4A,B showing the quantification of cytosolic PLA dots, which were not significantly altered upon A23187 treatment

Referee #1:

In this manuscript, the authors Ulferts et al, are dissecting the relationship between calcium-triggered nuclear actin polymerization and the nucleoskeleton, and what the functional consequences of disrupting this are on transcription. They find that SUN2 is required for nuclear f-actin assembly, and that this function is independent of SUN2's role in the LINC complex, that SUN2 and INF2 colocalize at the inner nuclear membrane in a calcium dependent manner, that SUN2 drives RNAP II clustering, and that this clustering is dependent on nuclear actin polymerization. The manuscript and figures are generally well organized and demonstrative of their data. Listed below are suggestions to improve the clarity and address some issues with the manuscript.

Major:

1. Line 89 - "actin filaments for euchromatin formation." It is not clear what you mean by this.
The phrase "actin filaments for euchromatin formation" was meant to convey the role of actin filaments in the structural organization and regulation of more open and dynamic chromatin (euchromatin). We have rephrased the sentence for clarity.
2. Line 90-92: What about INF2 is associated with disease? Please be more explicit about how INF2 (i.e. mutations within the protein, abnormal expression, dysfunction) contributes to disease pathologies.
We agree and we now added a sentence to specify the potential role of INF2 mutations in its dysregulation and pathophysiology.
3. Line 100 - " which receptors" - it is unclear what you are referring to by this.
We have edited the sentence accordingly.
4. Line 102-105 - the conclusions of the paper are slightly overstated. INF2 is never tested in the clustering of RNA Pol II. Further the statement of "disease-associated formin INF2" makes it seem you are testing a disease mutation and its role in this process.
We thank the reviewer for the comment and agree to remove the "disease-associated" attribute as we do not focus on any disease-associated mutation of INF2. However, thanks to the reviewer's suggestion we now performed the experiment testing the role of INF2 for RNA

Pol II cluster formation (Fig. 5). These results now show that INF2 plays a role in RNA Pol II clustering. We thank the reviewers for this suggestion.

5. Line 109-113. The rationale behind the experiments are not fully clear. In particular, the second sentence is hard to follow.

We have revised the sentence to better emphasize the rationale behind the experiments. Now line 117-118.

6. In the first paragraph of the Results it is important to explain the use of A23187

We added a sentence explaining the use of the calcium ionophore A23187. See line 123-124.

7. Figure 1 - it is not clear what you are measuring when you say nuclear F-actin (% of cells). A description is needed in the methods (and maybe the figure legend). It would also be clearer if the axis label was % of cells with nuclear F-actin.

A description of the quantification was and is provided in the methods section "Live cell spinning disc confocal microscopy and quantification" (line 377-383):
"Images were acquired with a Yokogawa CSU-X1 spinning disc using a 100x/1.46 oil objective and Photometrics Prime BSI camera and drugs (A23187 or thrombin) were added to the cells at the microscope while scanning. Time-lapse images were recorded for up to 4 min, and the cells displaying nuclear F-actin assembly were scored as positive. Conversely, cells lacking nuclear actin structures were classified as negative. To quantify these events, the number of positive occurrences was divided by the total cell count within the field of view, expressed as a percentage of cells."

8. Specificity of the INF2 antibody for immunofluorescence is not shown.

We thank the reviewer for the valuable feedback. We have now added a panel showing specificity of the INF2 antibody in siRNA depleted cells in Figure EV3.

9. Figure 2 - zoom in images of single nuclei would make it easier on the reader to see the differences.

Unfortunately, we believe that the reviewer may have confused Figure 2, as it already displays zoom-in images of the relevant structures (rather than individual nuclei). However, we follow his argument regarding Figure 3 and have added corresponding zoom-ins of individual nuclei there.

10. It would be easier to compare the data in figures 4 and 5 if in Fig 4 the images were next to each other and in both Fig. 4 and 5 the data were graphed on the same graph (ex. Control and siSUN2). Also the graph labels could be clearer - stating "number of" and "Percentage of" for current 4C/D, G/H and similarly in Fig 5.

We thank the reviewer for the comment and agree that reordering the figure will enhance clarity and facilitate better comparison of the treatment groups. We have therefore moved the example images of the siSUN2-treated cells up right below the control group. However, we believe that combining the data for both groups into a single graph would make it too cluttered. Instead, we have placed the corresponding graphs on top of each other to make data comparison easier.

We have also changed the graph labels according to the reviewer's suggestion.

11. In Figure 5, as a reader, I prefer to see the images in addition to the graphs of the quantification. I recommend showing just two time points.

We have added representative images to complement the graphs. Now Figure 6.

12. It is surprising the NLS-actin experiment was done in Figure 5 and not siINF2. The rationale for this change is not clear. Also the data for Figure 5 could be better explained in the results.

We have performed additional experiments and now added data on the role of INF2 on RNA Pol II clustering, which complement the previously presented data involving the non-polymerizable actin mutant (See Figure 5).

13. The conclusions of the paper could be strengthened by showing actin filaments associated with the RNA Pol II clusters and testing the role of INF2 in the clustering. Further, showing additional evidence the RNA Pol II clusters are actively transcribing would strengthen the paper.

Fixation of these nuclear actin filaments is, unfortunately, very challenging due to their highly transient nature. Nevertheless, we have tried but the glutaraldehyde-based fixation step needed for nuclear phalloidin staining (Baarlink et al. 2013; Krippner et al. 2020) interfered with the functionality of our RNA Pol II antibody.

As mentioned above, we tested the role of INF2 in the observed cluster formation.

We agree with the rationale of the reviewer and also believe that showing evidence for actively transcribing RNA Pol II clusters would strengthen the paper. However, establishing a method for nascent RNA imaging, such as through pulse-incorporation of EU and click chemistry-based conjugation of dyes (see Wei et al. 2020), or utilizing RNA FISH, is beyond the scope of the three-month time period allocated for revision, which we actually already exceeded. However, we will surely pursue this avenue further in the future.

14. Lines 172-174 - conclusion is slightly overstated. INF2 is never tested in RNA Pol II clustering, nor are the actin filaments shown to directly interact with RNA Pol II.

We have now included experimental data on the role of INF2 in RNA Pol II clustering (see Figure 5). Although we were unable to visualize actin filaments alongside RNA Pol II clusters, we believe that the experimental evidence presented, including the involvement of the actin nucleator INF2 and polymerizable-competent actin, further supports our conclusion.

Minor:

1. Line 78 - change "site" to "side" to illustrate where the nesprins connect to the nuclear envelope. "Site" makes it seem like there is only one specific point of connection whereas "side" makes it clearer that the connection can happen anywhere along the nuclear envelope.

The change has been made accordingly (see line 78).

2. Line 85 - add signaling after serum response factor.

"Signaling" has been added to the respective sentence. Now line 86.

3. Line 87 and elsewhere - "we could previously show" is odd phrasing. Recommend changing to "we previously showed/found"

We made the changes according to the reviewer's suggestion.

4. Line 115 - coma needed "SUN2 like INF2" to "SUN2, like INF2"

We have added a coma. Now line 125.

5. Line 129 - Figure callout is wrong. (Fig 2C) should be (Fig 2B)

6. Line 133 - Figure callout is wrong. (Fig 3C) should be (Fig 2C)

We thank the reviewer for the hint and have adjusted the references accordingly.

7. Line 165-167 - seems like this sentence belongs in the prior paragraph and this paragraph should start with the NLS-actin.

We agree with the reviewer and have added the sentence to the preceding paragraph. See line 180-183.

8. Line 196 - what is "the formin" referring to? INF2?

We thank the reviewer for the comment and have now rephrased the part for clarity. See line 214.

9. Line 207 - it is unclear what you mean by "autonomously"
We have rephrased this sentence now. See line 226.
10. The model could be better walked through in the discussion. Also, while KASH domains are not required for the SUN2 functions in the paper, they could still be bound to the SUN2 involved. This should be stated in the model figure legend.
We added a few sentences to guide through the proposed working model. Also, in the figure legends we put "the monomeric form of" into parenthesis
11. The calcium elevation using A23187 is not described in the methods
We have added a sentence in the methods section to describe the use of A23187. See line 277-279.
12. It is unclear if the quantifications were performed in a genotypically blinded manner.
Quantifications were not performed in a blinded manner.
13. Figure 1 - change labeling of westerns to be F, and make graph E.
We have included new data in Figure 1 and changed the labeling according to their appearance in the results part of the manuscript.
14. Figure 1 - why are the statistics used in 1C and 1F different.
All data sets were tested for normality before testing for a statistically significant difference of the treatment groups. While the data in Fig. 1C were normally distributed, a t-test was applied, whereas data in 1F were not and thus a Mann-Whitney-test was performed.
15. Figure 1B - referring to the color of the components you are talking about in the legend would help make it clearer.
We agree with the reviewer and have made the corresponding changes to the figure legend.
16. Figure 2C - were the IPs done in high calcium or not?
The IPs were performed in calcium containing buffer without EDTA as described in the methods section. See line 340-342.
17. I am surprised the full western blots are not provided as supplemental figures. It would also be nice if the raw data for the quantifications were provided as a supplemental file.
We apologize for not including the full Western blots for the initial submission. For the resubmission of the manuscript, we now provide all the raw data corresponding to the results presented in the main figures.

Referee #2:

This work follows up on a previous study by these authors, showing that the formin INF2 can cause actin polymerization in the nucleus in response to stimuli that elevate calcium, in addition to its known role in cytoplasmic actin polymerization. This INF2-mediated actin causes changes in chromatin organization. In the present manuscript, the authors show that the inner nuclear membrane (INM) protein SUN2 is necessary for this response to increased cytoplasmic actin, possibly through interaction with INF2. The actin polymerization is very rapid, peaking within 1 min of stimulus and dissipating within 2 min or so. The authors show that calcium stimulation also causes an increase in active clusters of RNA polymerase II, in a manner that is dependent on SUN2.

This work provides an interesting extension to the growing list of functions performed by actin in the nucleus. In addition, by tying SUN2 to the process the works links the actin to the INM. The experiments as presented are well controlled in general. The work, however, is quite 'thin', not delving into any of the results shown in any detail, and falling short on providing real insight. Additional experiments to raise significance would be useful.

Specific Comments:

1) In Figure 1, the authors quantify the response to A23187 treatment by % cells demonstrating nuclear actin. It would be useful to have a more detailed characterization of the response. For example, in Figure 1D, the siRNA treatment of SUN2 or INF2 results in a decreased percentage from ~65 to ~35. Is the actin response of the cells that still were classified as "positive" weaker than in siCtrl? Such an analysis would be more informative than what is given at present.

We thank the reviewer for the suggestion. However, the live imaging of the calcium mediated response is already extremely challenging due to the very narrow time window, allowing only for single-plane imaging in our setup. It is currently rather difficult, given our equipment, to robustly and reliably make out differences and establish a threshold for scoring a "weaker" or "stronger" response. While we agree that a more detailed quantification could be potentially interesting, we opted for a more straightforward yes/no quantification, similar to the approach used in the previously published study (Wang et al. 2019).

2) Figure 1C and Figure 1F. These are negative results for emerin and LINC effects on actin polymerization. It would be useful to have data showing that the treatments (emerin siRNA and dominant-negative KASH construct) are affecting known processes. Also, it would be useful to have some indication of the expression level of the dominant-negative KASH construct (western blot or cell staining).

We thank the reviewer for the suggestion. While it would have been beneficial if the reviewer had provided specific examples of processes that could be influenced by the corresponding treatments, we decided to perform immunofluorescence staining for endogenous nesprins in dnKASH expressing cells showing the displacement of the endogenous protein from the INM (see. Fig. EV1). This clearly demonstrated that overexpression of dnKASH does indeed effectively uncouple the SUN proteins from the endogenous nesprin proteins, thus interfering with the mechanotransduction process of the LINC complex. Unfortunately, we were only able to obtain specific antibodies for nesprin 2 and 3.

Expression of dnKASH in NIH3T3 cells is shown in Fig. EV2 (previously Fig. EV1).

3) It is not clear that the authors really 'prove that SUN2 is needed for nuclear actin polymerization independently of its role as a component of LINC. While the dominant-negative KASH construct is a start, additional experimentation (for example, siRNA of nesprins) would strengthen the result.

We thank the reviewer for the suggestion. We have added experimental data on the effect of Nesprin knockdown on calcium-induced nuclear actin assembly (see Fig. 1H), and we believe these results further strengthen the conclusion made in the manuscript. We are grateful for pointing this out.

4) Figure 2A and 2B show imaging of SUN2 and INF2 on the nuclear membrane, but that the two do not necessarily co-localize there, with SUN2 being quite homogenous whereas INF2 is somewhat punctate. It is, therefore, unclear what the significance of this figure is, considering that the authors' previous paper (reference 16) shows this localization for INF2, and that SUN2 is well known to localize to the INM. Given the finding that calcium causes actin polymerization dependent on both SUN2 and INF2, it would be interesting to test whether INF2 and/or SUN2 distribution changes on the same time scale as the actin polymerization. This experiment would be an important addition to the

PLA assays done subsequently, to show if the entire population of INF2 or SUN2 is changing following calcium addition, or whether it just appears to be a small percentage of the population.

We are grateful to the reviewer for this excellent idea. We performed IF staining of endogenous SUN2 and INF2 in untreated and A23187-treated cells and measured the INF2 fluorescence intensity within a 3D-rendered mask of the SUN2 “ring”. Interestingly, we found that the INF2 fluorescence intensity indeed increased upon calcium ionophore treatment suggesting that at least a pool of the INF2 protein redistributes towards the nuclear membrane upon calcium signaling. We have now included these novel results in Fig. 2C.

5) Figure 2C shows co-IP of SUN2 and INF2. Comments:

a. In the methods, the authors state that the cells were transfected with hSUN2-FLAG or INF2-CAAX separately. Does this mean that the INF2 they are detecting is transfected INF2, or endogenous INF2? If transfected, that construct does not appear to be listed.

We thank the reviewer for the suggestion and have added the missing plasmid information to the Materials and Methods section.

b. a useful negative control would be IP of emerin, another INM protein. Immunoprecipitations of membrane proteins can reveal a variety of non-specific interactions, depending on how well the membranes were detergent extracted.

We thank the reviewer for the suggestion. Through super-resolution imaging of cells co-stained for endogenous SUN2/INF2 as well as proximity ligation assays also assessing endogenous protein, we have demonstrated the likely close proximity of these proteins *in cellulo*. We merely performed the IP to support our endogenous colocalization data, however, we do not claim that this interaction is direct or independent of any other proteins. In our experiment, we used SUN2-FLAG as the bait protein. Since SUN2 and emerin are known interactors at the INM (see Haque F et al. J Biol Chem. 2010;285(5):3487-3498), we are unsure if emerin would be an ideal negative control. Identifying an appropriate INM protein as a negative control is challenging, so we propose moving the IP data to the supplementary material, since our focus remains on the imaging-based characterization of these proteins in an endogenous as well as more physiological setting.

c. Another important negative control would be to test co-IP of SUN2-FLAG with the non-ER bound isoform of INF2 (the isoform that lacks the CAAX box). If ER binding by INF2 is necessary for the interaction, the CAAX-less isoform should not bind.

Please see above, we do not claim any physical ‘interaction’ for both proteins. We apologize if this was unclear, and we now mention ‘associate’ or ‘colocalize’ in the manuscript text in order to make that more apparent. Further to that, we now write in line 211-212 that our data support a close vicinity but do not evidence a direct physical interaction, which of course would require extensive *in vitro* analysis using purified proteins.

6) Figure 3 shows PLA assays demonstrating SUN2/INF2 interaction. There are a number of comments on this analysis:

a. The examples shown in panel A are processed such that the PLA signal is weak. Despite this processing, there are clearly a number of PLA dots in the cytoplasm. The authors should be more transparent about this, by displaying brighter images, and commenting on the background cytoplasmic dots. Do these also change with A23187 treatment?

We thank the reviewer for this comment and did not mean to hide any of the unspecific PLA dots in the cytosol. We there provided brighter images as suggested. We also included a quantification of the detected cytosolic PLA dots, which were not significantly altered upon A23187 treatment (see Fig. EV4 A,B).

b. The authors supply some excellent negative controls, but another very useful negative control

would be to use another nucleoskeleton protein (such as emerin or lamin) to show that its interactions with INF2 are minimal or do not change with calcium addition.

We would kindly disagree here and we genuinely believe that we carefully executed our work, ensuring that all necessary negative and positive controls were included and presented in the supplementary files. Again, we also do not claim any physical 'interaction' for both proteins.

c. Also in Figure 3, the authors quantify the number of PLA dots, but there seems also to be a change in dot brightness. Quantification on brightness per dot would be a useful added parameter.

We thank the reviewer for the suggestion. However, we find the more stringent cut-off to be more conclusive, and we do not believe that quantifying brightness would significantly contribute additional critical information.

d. It would be useful to determine whether PLA dots are actin-dependent by treating with LataA.

Based on our experimental data, we conclude that the actin filaments are a downstream consequence of SUN2 and INF2 at the INM. In fact, our data demonstrate an actin-dependent clustering of RNA Pol II that is mediated by SUN2 and INF2 (see. Figs. 4 and 6).

7) Pertaining to the Pol2pS5 clusters shown in Figure 4:

a. The formation of large Pol2pS5 clusters in as little as 5 min post-calcium increase is fascinating. However, the change in nuclear actin takes place more rapidly, and is essentially gone by 5 min (movie 1). Is it possible for the authors to test Pol2pS5 clusters at earlier time points (such as 1 min post-stimulus)?

While we agree with the reviewer and find the formation of these RNA Pol II clusters within as little as 5 min very fascinating, we currently do not see the need to extend this time limit further. Additionally, conducting such experiments across all treatment groups would require extensive imaging and analysis that would take several months if not longer.

b. It is unclear whether the formation of large Pol2pS5 clusters is dependent on INF2. Testing the siINF2 cells for an effect on these clusters is an important control.

We agree with the reviewer and have added data on the role of INF2 on Pol2 clustering, which complement the previously shown data involving the non-polymerizable actin mutant.

8) In the Introduction, the authors state that INF2 exists mostly in a CAAX-box modified isoform that localizes to the ER membrane (ref 19. Chhabra et al 2009). Is this true for every cell type? Is the CAAX-lacking isoform more abundant in certain situations?

A study from 2011 suggests, that that the two isoforms are indeed expressed in a cell-type specific manner, with INF2-CAAX being predominant in NIH3T3 fibroblasts and the cytosolic non-CAAX being dominant in U2OS, HeLa, and Jurkat cells with both isoforms exerting distinct cellular functions (Ramabhadran et al. 2011). We thank the reviewer for their insightful comment and have now added a few sentences when introducing the INF2 protein in lines 96-99.

Referee #3:

Summary:

This manuscript explores the often overlooked role of the nuclear cytoskeleton in signal transduction of pro-transcriptional cues. The authors use calcium signaling as a potent model for exploring the mechanism of signaling to produce consistent, well supported observations for their experiments. A variety of imaging techniques are applied to answer specific questions about which proteins are involved in this process, and how they change their localization in response to increased intracellular

calcium levels. The results directly implicate SUN2 as a primary transducer of calcium signaling, independent of the force sensing pathway governed by KASH domain containing proteins. Based on the observation that the population of elongating RNA polymerase II increases cluster size during successful calcium signaling transduction, the authors conclude that calcium signaling is converted into a transcriptional response via the reorganization of nucleoskeleton components.

Key Findings:

The key finding described by the manuscript is that RNA polymerase clustering (inferred as transcription) is directly linked to the formation of nuclear actin filaments. I consider this a significant finding, but it could be supported by the additional suggested experiments (see below). As nuclear actin continues to be a difficult to study and enigmatic topic, such mechanistic work tying the phenomenon to a critical aspect of cell biology (transcription) is likely to be of wide interest to the community. The findings are well supported by the data, I find myself particularly convinced by the use of the dominant negative forms of actin abolishing the formation of RNA polymerase clusters. The confirmation of the RNAi knockdowns with western blots is also greatly appreciated.

Major Comments:

-The model put forth by the authors is easily interpreted and supported by strong experimental evidence. Specifically, the protein components implicated in the transduction of the actin signal are thoroughly investigated, through both the use of small molecule inhibitors and siRNA knockdown experiments. However, additional interrogation about the nature of the transcriptional response would increase the impact of the publication:

(1) I am interested in the role of actin polymerization instructing RNA polymerase clustering. Specifically, would it be possible to see if actin polymerization co-localizes with the RNA polymerase clusters that appear upon calcium signaling induction? Given the reagents on hand (RNA Pol II pSer5 antibody, actin chromobody), this should be relatively straightforward to perform as an immunostain, although there might be additional technical limitations that I have not considered.

We thank the reviewer for the suggestion and agree that the visualization of RNA Pol II cluster at/close to nuclear actin F-actin would strengthen our argumentation. However, fixation of these nuclear actin filaments is challenging due to their very transient nature. Nevertheless, we have tried to do so but found that the glutaraldehyde-based fixation step needed for nuclear phalloidin staining (Baarlink et al. 2013; Krippner et al. 2020) interfered with the functionality of our RNA Pol II antibody. We therefore are unable at present to pursue this approach further.

(2) Is RNA polymerase clustering induced on genes that are expressed in a response to A23187 treatment? Combining an RNA FISH stain for a A23187 responsive gene with the RNA Pol II antibody stain would directly associate the newly formed clusters with the induced transcription. This is a much larger ask since it would be the development of a new assay, and therefore I would not consider it strictly necessary. However, I would say that explicitly stating that this reorganization of Pol II is indicative of transcription (Line 102) should be tempered if this evidence is not provided.

We agree with the reviewer and also believe that showing evidence for actively transcribing RNA Pol II clusters would strengthen the paper. However, establishing a method for nascent RNA imaging, such as through pulse-incorporation of EU and click chemistry-based conjugation of dyes as demonstrated by Wei et al. (Science Advances 2020), or utilizing RNA FISH, is beyond the scope of the three-month time period allocated for revision. We followed the reviewer's suggestion and have therefore rephrased the sentence (see line 110-111).

-As mentioned before, I am impressed by the dominant negative form of actin preventing the formation of RNA polymerase clustering. I understand that this perturbation is well known in the

actin field, but if it is feasible I believe it would bolster the findings to perform the actin chromobody experiment on cells expressing this construct to demonstrate that preventing polymerization of the nuclear actin filaments is the likely mechanism of action.

We thank the reviewer for their valuable feedback. The non-polymerizable ActinR62D mutant is indeed well-established and a widely used tool in the nuclear actin research field given the limited choice of interfering with actin dynamics more specifically in the nuclear compartment only. Nevertheless, our previous studies, have already established that the nuclear-localized NLS-actin-R62D effectively disrupts nuclear actin polymerization and its downstream effects in NIH3T3 cells used here as well (see Baarlink et al. 2013 Fig. S1D and Baarlink et al. 2017 Fig. 3e).

Minor Comments:

-Does the actin polymerization happen throughout the nucleus, or is it restricted to the nuclear lamina? Perhaps the authors could include a XZ view of the actin chromobody fluorescence (like the projections used for the PLA experiments) to visualize this.

We thank the reviewer for the suggestion. However, all live imaging performed for quantifying nuclear F-actin events was conducted using single-plane imaging. Given the rapid and dynamic nature of the calcium-induced response, performing 3D imaging is at present not possible with our setup. This will surely be an important future task.

-For supplemental figure EV1, it is difficult to see the actin polymerization in response to the drug. If the images were enlarged like they are in Figure 1, it would help avoid confusion.

We agree and thank the reviewer for the suggestion and have now included zoom-in images of individual nuclei to provide better images nuclear F-actin formation (see now Fig. EV2).

-While it is very much appreciated that the authors display their data as violin plots with individual replicates labeled as different colors, it is a bit visually distracting for the viewer. The authors may want to simplify the plotting style and show more complete plots in the supplemental materials, or simply reduce the opacity of the individual data points and emphasize the horizontal lines that denote the median and quartile values.

We agree and thank the reviewer for the suggestion and have reduced the opacity of the individual data points of the corresponding graphs. We additionally emphasized the horizontal lines denoting the median.

References

- Baarlink C, Plessner M, Sherrard A, Morita K, Misu S, Virant D, Kleinschnitz EM, Harniman R, Alibhai D, Baumeister S, et al. 2017. A transient pool of nuclear F-actin at mitotic exit controls chromatin organization. *Nat Cell Biol* **19**: 1389–1399.
- Baarlink C, Wang H, Grosse R. 2013. Nuclear actin network assembly by formins regulates the SRF coactivator MAL. *Science (80-)* **340**: 864–867.
- Krippner S, Winkelmeier J, Knerr J, Brandt DT, Virant D, Schwan C, Endesfelder U, Grosse R. 2020. Postmitotic expansion of cell nuclei requires nuclear actin filament bundling by α -actinin 4. *EMBO Rep* **21**: e50758. <https://onlinelibrary.wiley.com/doi/full/10.15252/embr.202050758> (Accessed November 9, 2022).
- Ramabhadran V, Korobova F, Rahme GJ, Higgs HN. 2011. Splice variant-specific cellular function of the formin INF2 in maintenance of Golgi architecture. *Mol Biol Cell* **22**: 4822–4833.
- Wang Y, Sherrard A, Zhao B, Melak M, Trautwein J, Kleinschnitz EM, Tsopoulidis N, Fackler OT, Schwan C, Grosse R. 2019. GPCR-induced calcium transients trigger nuclear actin assembly for chromatin dynamics. *Nat Commun* **10**: 1–9. <http://dx.doi.org/10.1038/s41467-019-13322-y>.
- Wei M, Fan X, Ding M, Li R, Shao S, Hou Y, Meng S, Tang F, Li C, Sun Y. 2020. Nuclear actin regulates inducible transcription by enhancing RNA polymerase II clustering. *Sci Adv* **6**.

Dear Robert,

Thank you for submitting your revised manuscript. It has now been seen by two of the original referees.

As you can see, the referees find that the study is significantly improved during revision and recommend publication. However, I need you to address the editorial points below before I can accept the manuscript.

- Please rename 'Data and materials availability' section as 'Data availability'. Please change the statement as 'This study includes no data deposited in external repositories.' and please move the section before Acknowledgments.
- Please rename 'Competing interests' section as 'Disclosure Statement and Competing Interests'.
- Please remove the 'Author contributions' section from the manuscript.
- As per our format requirements, in the reference list, citations should be listed in alphabetical order and then chronologically, with the authors' surnames and initials inverted; where there are more than 10 authors on a paper, 10 will be listed, followed by 'et al.'. Please see <https://www.embopress.org/page/journal/14693178/authorguide#referencesformat>. References need to be alphabetical (not numerical), et al should be used after 10 author names, DOI should only be used for preprints and datasets that have not been published yet.
- Please remove the reference to BioRender from the figure legends. Instead, please add this information the Methods section as a separate subheading at the very end as follows: "Graphics. BioRender was used to prepare the Figures 1B and 7 and the synopsis image of this study."
- Please include the funding information in the Acknowledgements section and remove the title Funding.
- We note that the panels of Fig. 5 (A-H) and Fig. 6 (A-H) are not individually called out in the text.
- We note the following regarding the movies: movie legends should be removed from the ms file; each legend should be provided in a readme.txt file and then it should be zipped up together with its movie so that we have one folder uploaded per one movie.
- All research articles submitted as revised versions must include a structured methods section that includes a Reagents and Tools Table followed by a Methods and Protocols section. Please see <https://www.embopress.org/page/journal/14693178/authorguide#structuredmethods> for further information.
- Materials and Methods should be Methods.
- Figures and Tables should be Figures (as there aren't any tables).
- Supplementary Material in the ms should be renamed to Expanded View Figure Legends and the title of each figure should be Figure EV1, etc. (instead of just EV1)
- Please re-check blots. Blots are empty in the figures but can see visible blots in the source data - i.e. Figure 1E - Sun2 / Figure 1F - Nesp1 2 and Nesp1 3 / Possibly also Figure EV3 3 - Flag (left blot).
- Our production/data editors have asked you to clarify several points in the figure legends:
 - o Please note that the exact p values are not provided in the legends of figures 1d; 3b, d; 4c-e; 5c-e; 6c, e.
 - o Please note that the box plots need to be defined in terms of minima, maxima, bounds of box and whiskers in the legend of figure 2c.
- Papers published in EMBO Reports include a 'synopsis' and 'bullet points' to further enhance discoverability. Both are displayed on the html version of the paper and are freely accessible to all readers. The synopsis includes a short standfirst summarizing the study in 1 or 2 sentences (max 35 words) that summarize the paper and are provided by the authors and streamlined by the handling editor. I would therefore ask you to include your synopsis blurb and 3-5 bullet points listing the key experimental findings.
- In addition, please provide an image for the synopsis. This image should provide a rapid overview of the question addressed in the study but still needs to be kept fairly modest since the image size cannot exceed 550 (width) x 300-600 (height) pixels.

Thank you again for giving us to consider your manuscript for EMBO Reports, I look forward to your minor revision.

Kind regards,

Deniz

--

Deniz Senyilmaz Tiebe, PhD
Senior Scientific Editor
EMBO Reports

Referee #1:

The revised manuscript is improved and addresses all my concerns.

Referee #3:

The authors have adequately addressed my concerns. I have no further comments to provide, and I believe the manuscript should be published.

All editorial and formatting issues were resolved by the authors.

Prof. Robert Grosse
Albert-Ludwigs-Universität Freiburg
Department of Pharmacology
Albertstraße 25
Freiburg 79104
Germany

Dear Robert,

Thank you for submitting your revised manuscript. I have now looked at everything and all is fine. Therefore, I am very pleased to accept your manuscript for publication in EMBO Reports.

Congratulations on a nice work!

Kind regards,

Deniz

--

Deniz Senyilmaz Tiebe, PhD
Senior Scientific Editor
EMBO Reports

--
